# Exploring Interpretability for Visual Prompt Tuning with Cross-layer Concepts

**Yubin Wang**[1,*] **Xinyang Jiang**[2,†]**, De Cheng**[3]**, Xiangqian Zhao**[3]**, Zilong Wang**[2]**,**
**Dongsheng Li**[2]**, Cairong Zhao**[1,†]
[1]School of Computer Science and Technology, Tongji University
[2]Microsoft Research Asia
[3]School of Telecommunication and Engineering, Xidian University

## Abstract

Visual prompt tuning offers significant advantages for adapting pre-trained visual foundation models to specific tasks. However, current research provides limited insight into the interpretability of this approach, which is essential for enhancing AI reliability and enabling AI-driven knowledge discovery. In this paper, rather than learning abstract prompt embeddings, we propose the first framework, named **I**nterpretable **V**isual **P**rompt **T**uning (IVPT), to explore interpretability for visual prompts by introducing cross-layer concept prototypes. Specifically, visual prompts are linked to human-understandable semantic concepts, represented as a set of category-agnostic prototypes, each corresponding to a specific region of the image. IVPT then aggregates features from these regions to generate interpretable prompts for multiple network layers, allowing the explanation of visual prompts at different network depths and semantic granularities. Comprehensive qualitative and quantitative evaluations on fine-grained classification benchmarks show its superior interpretability and performance over visual prompt tuning methods and existing interpretable methods. Our code is available at `https://github.com/ThomasWangY/IVPT`.

## 1 Introduction

Visual prompt tuning (Jia et al., 2022) has emerged as a promising approach for adapting pre-trained visual foundation models (He et al., 2022; Chen et al., 2021; Dosovitskiy, 2020) to specific tasks, allowing for flexible task customization while avoiding the need for full-scale model fine-tuning. This approach has shown significant advantages in terms of efficiency and task adaptability, yet it still faces a major challenge in interpretability. Most prompt tuning techniques (Jia et al., 2022; Dong et al., 2022; Gao et al., 2022) involve learning abstract embeddings that automatically capture high-level features, providing implicit guidance for the model but offering limited human-understandable information into its decision-making process. The lack of transparency in these prompts hampers the ability to assess the trustworthiness of AI systems and limits the scope for uncovering valuable insights through AI-driven analysis, especially in safety-critical domains such as healthcare and autonomous driving. Although existing multi-modal approaches (Bie et al., 2024; Yao et al., 2023; Bulat & Tzimiropoulos, 2023) seek to improve prompt interpretability using human-designed guidance, such as natural language, autonomously discovering clear and meaningful explanations for visual prompts during training remains a significant challenge.

Recent methods have been developed to enhance the interpretability of visual models, such as concept-based methods (Fel et al., 2023; Ghorbani et al., 2019; Kim et al., 2018; Zhang et al., 2021) that utilize high-level abstractions to represent learned features and attribution-based methods (Yosinski et al., 2015; Selvaraju et al., 2017) that identify regions in the input image that significantly influence the model's predictions. In this paper, we focus on an interpretable scheme that integrates both attribution and concept discovery. We aim to interpret abstract visual prompts by linking them to human-understandable concepts, each grounded in distinct image regions and assigned an importance

---

*This work was done during Yubin Wang's internship at Microsoft Research Asia
†Corresponding Authors. Email: xinyangjiang@microsoft.com, zhaocairong@tongji.edu.cn

score based on its contribution to the model's prediction, all achieved in an unsupervised manner, without relying on annotations or internal model access. For example, given an image of a bird, we learn an interpretable prompt token with the concept of a "bird wing", quantifies its impact on predicting the class "bird", and localizes this concept to the wing region in the image.

Recently, several methods (Chen et al., 2019; Nauta et al., 2021; Wang et al., 2021; Huang et al., 2023) have explored this attribution-concept hybrid interpretability scheme by learning part prototypes to represent specific concepts. However, these methods are primarily designed for conventional neural network architectures rather than visual prompt tuning, presenting three key challenges. First, prior methods focus on grounding concepts to image regions, while the connection between these concepts and abstract prompt embeddings remains largely unexplored. Second, existing methods extract concepts from the features of the final layer of deep models, neglecting the need to interpret visual prompts learned at different layers and to capture cross-layer interactions among concepts. Third, existing approaches typically learn a separate set of prototypes for each class, making it difficult to analyze the model's behavior across classes. We may be unable to capture and interpret shared concepts that may appear in multiple categories. We assume that prototypes are not inherently tied to specific pixels but gain semantic meaning through their similarity to localized patches in the image.

As a result, we introduce **I**nterpretable **V**isual **P**rompt **T**uning (IVPT), a novel framework that emphasizes the interpretability of visual prompt tuning. IVPT advances beyond abstract embedding learning by introducing cross-layer concept prototypes that connect learnable prompts with human-understandable visual concepts. Specifically, each interpretable prompt is generated by aggregating features from an image region corresponding to a concept prototype at a specific layer. These prototypes are distributed across multiple network layers, enabling IVPT to interpret visual prompts at different semantic depths. Unlike traditional methods, IVPT learns category-agnostic concept prototypes, enabling the model to capture shared, non-overlapping concepts across various categories, offering a more coherent explanation by focusing on the commonality of concepts. We also observe that coarse-grained prompts in deep layers capture high-level concepts but lose fine-grained details, while overly specific prompts in shallow layers lack broader contextual understanding. To capture interactions between concepts across layers, IVPT bridges the gap with cross-layer prompt fusion to align fine- and coarse-grained tokens. By linking local-to-global semantics across network layers with varying granularity, IVPT emulates human visual reasoning, thus enhancing interpretability.

The main contributions of this paper are threefold: (1) We propose a novel framework for interpretable visual prompt tuning that uses concept prototypes as a bridge to connect learnable prompts with human-understandable visual concepts. (2) We introduce cross-layer concept prototypes to explain prompts at multiple network layers while modeling their relationships in a fine-to-coarse alignment. (3) We demonstrate the effectiveness of our approach through extensive qualitative and quantitative evaluations on fine-grained classification benchmarks and pathological images, showing improved interpretability and accuracy compared to both conventional VPT methods and interpretable methods.

## 2 RELATED WORKS

### 2.1 INTERPRETABILITY OF VISUAL MODELS

Numerous methods have been developed to enhance the interpretability of visual models. Concept-based methods (Fel et al., 2023; Zhang et al., 2021) use high-level abstractions to represent learned features, facilitating understanding of the internal representations of a model. However, these approaches typically operate at a single layer and lack region-level grounding, failing to provide mechanisms for linking concepts to prompt embeddings or enabling multi-layer interpretability. Attribution-based methods (Yosinski et al., 2015; Selvaraju et al., 2017) identify key regions in the input images that affect predictions, clarifying the focus of the model during making decisions. While effective at identifying influential regions, these methods do not offer semantic concept alignment or interpretability of prompt tokens, especially across multiple layers. However, these approaches often lack transparency when applied to visual prompt tuning and generally fail to provide fine-grained, hierarchically organized explanations. To address this issue, we focus on part-prototype methods (Chen et al., 2019; Nauta et al., 2021; Wang et al., 2021; Huang et al., 2023), which offer localized representations associated with specific image regions. ProtoPNet (Chen et al., 2019), as the foundational model, compares input features with prototypes of object parts, while its extensions (Rymarczyk et al., 2021; 2022; Ma et al., 2024) further enhance interpretability.

ProtoTree (Nauta et al., 2021) integrates prototypes with decision trees, while TesNet (Wang et al., 2021) arranges prototypes for spatial regularization. Further advancements include evaluating prototype interpretability using a benchmark developed by Huang et al. (Huang et al., 2023). However, these frameworks face limitations when applied to visual prompt tuning: they (1) lack concept-prompt linkage, unable to connect concepts to prompt embeddings; (2) lack cross-layer interpretation, being restricted to final-layer prototypes; and (3) rely on class-specific concepts, limiting cross-category analysis. Consequently, they neither define trainable prompts nor organize concepts hierarchically across layers. In contrast, IVPT addresses this gap by employing category-agnostic prototypes at multiple layers to define interpretable prompts and model their fine-to-coarse relationships.

## 2.2 VISUAL PROMPT TUNING

Visual prompt tuning (Bowman et al., 2023; Hu et al., 2022; Loedeman et al., 2022; Wu et al., 2022; Jia et al., 2022) has become a popular parameter-efficient approach to transfer the generalization capabilities of pre-trained vision models to various downstream tasks. Recent studies (Dong et al., 2022; Zhao et al., 2024; Wang et al., 2024; Sohn et al., 2023; Zheng et al., 2025) focus on modifying inputs by embedding learnable parameters, aiming to adjust the input distribution and enable frozen models to handle new tasks. For instance, VPT (Jia et al., 2022) introduces a limited set of learnable parameters as input tokens to the Transformer. PViT (Herzig et al., 2024) uses specialized parameters in a shared video Transformer backbone for both synthetic and real video tasks. E$^2$VPT (Han et al., 2023) incorporates learnable key-value prompts in self-attention layers and visual prompts in input layers to improve fine-tuning. Gated Prompt Tuning (Yoo et al., 2023) learns a gate for each ViT block to adjust its intervention into the prompt tokens. Despite their exceptional performance, these methods fundamentally treat prompts as unconstrained black-box vectors, providing no inherent interpretability or semantic grounding. They lack three critical properties: (i) spatial grounding to concrete image regions, (ii) human-understandable concept definition, and (iii) cross-layer semantic structure. This represents a significant gap in the VPT literature, as interpretability remains limited to final features/logits rather than the prompts themselves. Prompt-CAM (Chowdhury et al., 2025) learns class-specific prompts for a pre-trained ViT and using the corresponding outputs for classification. However, these learned prompt tokens are typically unconstrained embedding vectors: they are not grounded in image regions, associated with reusable concepts, or linked across layers into a coherent hierarchy. In contrast, IVPT redefines prompts as region-grounded, category-shared concept embeddings that are explicitly aligned from shallow attributes to deeper parts across layers. Thereby, our work establishes the first interpretable paradigm for VPT—a concept-attribution hybrid—enabling direct semantic interpretation of prompts while maintaining parameter efficiency.

## 3 IVPT: INTERPRETABLE VISUAL PROMPT TUNING

### 3.1 OVERALL PIPELINE

In this subsection, we explain how the proposed IVPT enhances the interpretability of visual prompts with the explainable properties of concept prototypes, as illustrated in Figure 1.

**Constructing interpretable prompts.** Given a pre-trained Transformer model of $N$ layers, we learn a set of continuous embeddings to serve as prompts within the input space of a Transformer layer, following the methodology outlined in Visual Prompt Tuning (Jia et al., 2022). Specifically, for $(i + 1)$-th Layer $L_{i+1}$, we denote the collection of $n$ input learnable prompts as $\mathbf{P}_i = \{\mathbf{p}_k^i \in \mathbb{R}^d \mid k \in \mathbb{N}, 1 \leq k \leq n\}$, where $d$ denotes the feature dimension of the ViT backbone. During fine-tuning, only the prompt embeddings are updated, while the Transformer backbone remains frozen.

However, the learned prompts are often abstract and difficult to interpret. To better explain their meaning, it is necessary to establish a way of relating them to specific human-understandable concepts. We use prototypes to convey a set of specific meanings. These prototypes are category-agnostic, generally focusing on particular image regions sharing similar semantics across images of different categories. As a result, a set of $m$ concept prototypes $\mathbf{Q} = \{\mathbf{q}_k \in \mathbb{R}^d \mid k \in \mathbb{N}, 1 \leq k \leq m\}$ is proposed to interpret a group of prompts $\mathbf{P}$ in a specific Transformer layer[1] as:

$$\mathbf{p}_k = \mathcal{F}(\mathbf{q}_k, \mathbf{E}), \quad k = 1, \ldots, m. \tag{1}$$

---

[1]The layer index $i$ is omitted in subsequent equations, as they represent computations for a specific layer.

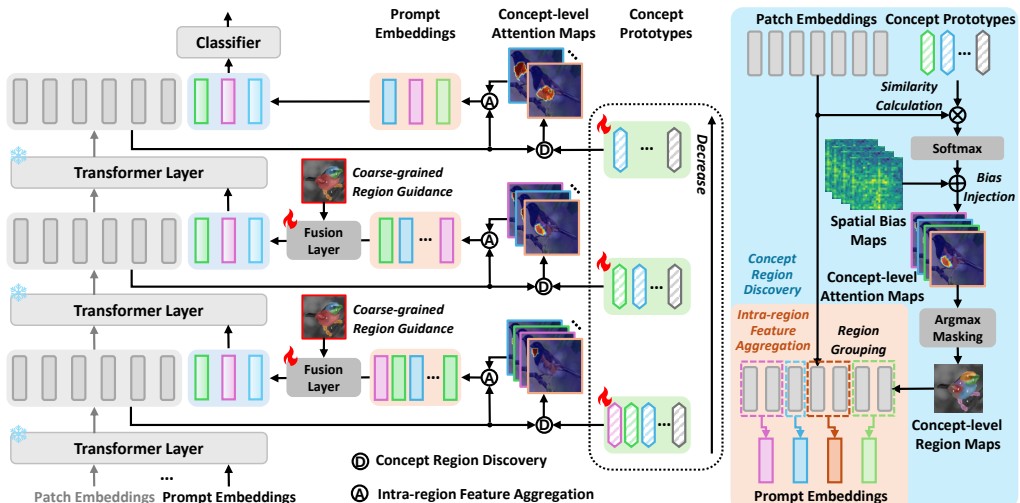

Figure 1: IVPT introduces category-agnostic concept prototypes to generate explainable visual prompt embeddings. At each layer, Concept Region Discovery (CRD) module captures specific visual concepts as concept-level image regions, while Intra-region Feature Aggregation (IFA) module aggregates features grouped by these region maps to obtain prompt embeddings. Several sets of layer-wise concept prototypes are used to capture concepts across layers, while the fusion layer, guided by the coarse-grained region, fuses the prompts in a fine-to-coarse manner as the final prompts to replace flat visual prompts at each layer.

Here $\mathbf{E} \in \mathbb{R}^{(h \times w) \times d}$ indicates the patch embeddings at the corresponding layer, where $h$ and $w$ represent the spatial dimensions (height and width), and $d$ is the feature dimension. In this context, the function $\mathcal{F}$ is learned in IVPT to map each prompt embedding $\mathbf{p}_k$ to a corresponding concept prototype $\mathbf{q}_k$, a process that will be further explained. To better understand the concept represented by a prototype $\mathbf{q}_k \in \mathbf{Q}$, IVPT grounds this prototype to a concept region $\mathbf{R}_k$ within a specific image by our proposed Concept Region Discovery (CRD) module, denoted as the function $\mathcal{F}_{CRD}$:

$$\mathbf{R}_k = \mathcal{F}_{CRD}(\mathbf{q}_k, \mathbf{E}), \quad k = 1, \ldots, m. \tag{2}$$

This operation explains the semantics of prototypes at the image level by highlighting the region with high attention on the image. Given a concept prototype $\mathbf{q}_k$, its corresponding prompt embedding $\mathbf{p}_k$ is derived with features within the concept region $\mathbf{R}_k$ through our designed Intra-region Feature Aggregation (IFA) module, denoted as the function $\mathcal{F}_{IFA}$:

$$\mathbf{p}_k = \mathcal{F}_{IFA}(\mathbf{R}_k, \mathbf{E}), \quad k = 1, \ldots, m. \tag{3}$$

Finally, we express the function $\mathcal{F}$ as $\mathcal{F} = \mathcal{F}_{CRD} \circ \mathcal{F}_{IFA}$, enabling us to present prompts with prototypes more interpretably. $\mathcal{F}$ in Eq. 1 consists of two steps, where CRD in Eq. 2 uses prototype $q_k$ as a semantic anchor to discover localized image regions $R_k$ and IFA in Eq. 3 aggregates token embedding $E$ within $R_k$ to obtain $p_k$. $m$ concept prototypes $q_k$ explain each visual prompt $p_k$ by grounding it in semantically meaningful image regions and representing it with the feature of these regions. More details will be provided in Section **Concept-prototype-based prompt learning**.

**Exploring cross-layer interpretation.** To interpret prompts across network layers with varying granularity, we propose cross-layer concept prototypes. At each layer, prompts are represented by a layer-specific set of prototypes, as shown in Figure 1. We adopt more prototypes in shallower layers to capture fine-grained and diverse visual features. As the network depth increases, the semantic representations become more abstract, and the number of prototypes progressively decreases, reflecting the increased conceptual coherence of features in deeper layers. Section **Cross-layer prompt fusion** will explain how IVPT fuses fine-grained prompts into $n$ prompts integrated for fine-tuning. Different from (Jia et al., 2022), we consider the prompts $\mathbf{P}_N$ after the last Transformer layer as the final concept representations, which are passed into a classification head to output individual concept-conditioned category scores $\mathbf{s}_k$. These scores reflect the likelihood of the input

image belonging to a given class, conditioned on the interpretable prompt $\mathbf{p}_k$, highlighting the significance of each concept for classification. The overall score is obtained by averaging individual concept-conditioned category scores, allowing the model to aggregate multiple complementary semantic perspectives for more robust predictions. We use cross-entropy to calculate the classification loss $\mathcal{L}_{cls}$ by comparing the score with the one-hot groundtruth label $\mathbf{y}$:

$$\mathcal{L}_{cls} = \text{CELoss}(\frac{1}{n}\sum_{k=1}^{n}\mathbf{s}_k, \mathbf{y}), \quad \text{where } \mathbf{s}_k = \text{Head}_k(\mathbf{p}_k^N). \tag{4}$$

This formulation follows the common practice in prototype-based interpretable models of aggregating multiple concept-conditioned category scores.

### 3.2 Concept-prototype-based prompt learning

In this subsection, we dive into the pipeline of learning explainable prompts with concept prototypes.

**Concept region discovery.** Concept Region Discovery (CRD) module associates concept prototypes with specific image regions, denoted as function $\mathcal{F}_{CRD}$ in Eq. 2. First, the patch embeddings $\mathbf{E}$ produced by the ViT are reshaped into a feature map $\tilde{\mathbf{E}} \in \mathbb{R}^{h \times w \times d}$. Then, we compute concept-level attention maps $\mathbf{A} \in [0, 1]^{m \times h \times w}$, with each element $a_{k,ij}$ representing the attention score of a learnable concept prototype embedding $\mathbf{q}_k \in \mathbf{Q}$ on the patch embedding $\mathbf{e}_{ij} \in \tilde{\mathbf{E}}$. Similarly to previous research (Aniraj et al., 2024; Huang & Li, 2020; Hung et al., 2019), we obtain the attention map using a negative squared Euclidean distance function to measure similarity, followed by a Softmax function across the $m$ channels and injection of spatial bias:

$$a_{k,ij} = \frac{\exp\left(-\|\mathbf{e}_{ij} - \mathbf{q}_k\|^2\right)}{\sum_{l=1}^{m}\exp\left(-\|\mathbf{e}_{ij} - \mathbf{q}_l\|^2\right)} + b_{k,ij}, \tag{5}$$

where $b_{k,ij}$ denotes the corresponding element in learnable spatial bias maps $\mathbf{B} \in \mathbb{R}^{m \times h \times w}$. We employ a combination of part-shaping loss functions from (Aniraj et al., 2024) to guide the discovery of non-overlapping regions associated with prototypes, denoted as $\mathcal{L}_{ps}$. This set of loss functions ensures the discovery of distinct, transformation-invariant parts with unique assignments, foreground presence, universal background, minimum connectivity, and minimize prototype polysemanticity, whose details are provided in the Appendix. Finally, concept region maps $\mathbf{R} \in [0, 1]^{m \times h \times w}$ are generated based on the attention map $\mathbf{A}$, where each image patch is assigned the attention value of the concept that has the highest attention score in $\mathbf{A}$. Thus, each element $r_{n,ij} \in \mathbf{R}$ represents the probability that a given image patch belongs to the corresponding concept and is defined as:

$$r_{n,ij} = \begin{cases} a_{n,ij} & \text{if } n = \arg\max_k a_{k,ij} \\ 0 & \text{otherwise.} \end{cases} \tag{6}$$

**Intra-region feature aggregation.** We introduce Intra-region Feature Aggregation (IFA) module corresponding to function $\mathcal{F}_{IFA}$ in Eq. 3, which obtains an interpretable prompt corresponding to a specific concept by aggregating the patch embeddings $\tilde{\mathbf{E}}$ within the corresponding concept regions. Specifically, we first calculate region-conditional feature maps $\mathbf{Z} \in \mathbb{R}^{m \times h \times w \times d}$, where the feature at each location is re-weighted based on the specific region and the probability element indicated in $\mathbf{R}$ by $\mathbf{Z} = \mathbf{R} \otimes \tilde{\mathbf{E}}$, where $\otimes$ denotes an unsqueeze operation on the last dimension of $\mathbf{R}$, followed by element-wise multiplication with $\tilde{\mathbf{E}}$. Finally, we aggregate features within each region to derive the prompt $\mathbf{p}_k$ corresponding to the $k$-th concept prototype:

$$\mathbf{p}_k = \frac{\sum_{i,j}\mathbf{z}_{k,ij}}{\sum_{i,j}r_{k,ij}}, \quad k = 1, 2, \ldots, m, \tag{7}$$

where $\mathbf{z}_{k,ij} \in \mathbb{R}^d$ represents a feature vector in $\mathbf{Z}$.

### 3.3 Cross-layer prompt fusion

IVPT employs cross-layer concept prototypes to represent prompts from different Transformer layers, capturing varying levels of semantic granularity. Shallow prompts are associated with

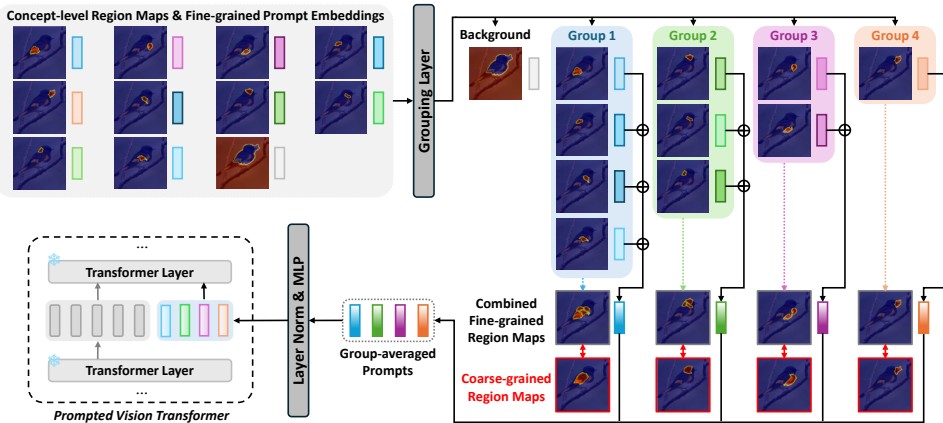

Figure 2: Illustration of cross-layer prompt fusion.

more prototypes reflecting fine, low-level semantics, while deep prompts are associated with fewer prototypes with coarse, high-level concepts. To explore relationships among prompts with different semantic granularities, IVPT introduces cross-layer prompt fusion, where prompt embeddings at shallow layers are combined to form deep prompts containing high-level concepts, as shown in Figure 2. Let $\mathbf{P}$ denotes the set of prompts generated in the previous section. We partition the prompts into groups based on shared high-level semantics by assigning each $\mathbf{p}_k$ a group label $y_k = f_g(\mathbf{p}_k)$, where $0 \le y_k \le n$. Here, $y_k = 0$ denotes the background class, and $n$ represents the total number of deep-layer prompts with distinct high-level semantics. The grouping layer $f_g$ is implemented as a linear layer followed by a Gumbel-Softmax (Jang et al., 2016) operation to find the group with the largest value. Within each group $i$, prompts are aggregated by computing their mean vector $\overline{\mathbf{p}}_i$, which is associated with a high-level concept.

To enable the group layer to effectively establish the intended correspondence between prompts in the shallow layers and those in the deep layers, we introduce a concept region consistency loss $\mathcal{L}_{con}$. This loss ensures that the combined concept regions associated with a group of fine-grained prompts align closely with a coarse-grained region from the final layer corresponding to a high-level concept. Formally, the concept region consistency loss is defined as the KL divergence between a set of grouped fine-grained region maps and their corresponding coarse-grained region maps:

$$\mathcal{L}_{con} = \frac{1}{n} \sum_{i=1}^{n} \text{KL}(\mathbf{R}_i^f, \mathbf{R}_i^c), \text{ where } \mathbf{R}_i^f = \sum_{k \in \mathcal{S}_i} \mathbf{R}_k. \tag{8}$$

Here $\mathbf{R}_i^f$ denote the combined fine-grained region map of the $i$-th group, and $\mathbf{R}_i^c$ is the $i$-th coarse-grained region map, which is derived from the last Transformer layer and used for classification. This alignment encourages the fine-grained prompts to merge cohesively, resulting in better alignment with the coarse-grained prompts with high-level semantics. The averaged prompts $\overline{\mathbf{p}}_i$ are then passed through layer normalization and an MLP:

$$\mathbf{p}_i^{fs} = \text{MLP}(\text{LayerNorm}(\overline{\mathbf{p}}_i)), \quad i = 1, \ldots, n. \tag{9}$$

The set of fused prompts $\mathbf{P}^{fs} = \{\mathbf{p}_i^{fs} \in \mathbb{R}^d \,|\, i \in \mathbb{N}, 1 \le i \le n\}$ are finally used to tune the network.

The total training loss is defined as:

$$\mathcal{L} = \lambda_{cls}\mathcal{L}_{cls} + \lambda_{ps}\mathcal{L}_{ps} + \lambda_{con}\mathcal{L}_{con}, \tag{10}$$

where $\lambda_{cls}$, $\lambda_{ps}$ and $\lambda_{con}$ are balancing ratios for these losses. Specifically, the part-shaping loss and the concept region consistency loss are averaged across layers. These losses enable our model to balance classification performance with interpretability, thus promoting explainable recognition.

Table 1: Quantitative comparisons of interpretability using consistency scores (Con.) and stability scores (Sta.), as well as the accuracy (Acc.) on the CUB-200-2011 (Wah et al., 2011) dataset are provided. The results of IVPT are compared against related approaches, including conventional part-prototype networks and various visual prompt tuning methods. Best in bold.

| Methods | DeiT-S | | | DeiT-B | | | DinoV2-S | | | DinoV2-B | | | DinoV2-L | | |
|---|---|---|---|---|---|---|---|---|---|---|---|---|---|---|---|
| | Con. | Sta. | Acc. | Con. | Sta. | Acc. | Con. | Sta. | Acc. | Con. | Sta. | Acc. | Con. | Sta. | Acc. |
| *Conventional Part-Prototype Networks* | | | | | | | | | | | | | | | |
| ProtoPNet (Chen et al., 2019) | 14.7 | 52.7 | 80.2 | 16.2 | 60.5 | 81.0 | 23.9 | 52.8 | 82.7 | 27.6 | 57.0 | 85.8 | 26.7 | 59.9 | 86.1 |
| ProtoPool (Rymarczyk et al., 2022) | 26.9 | 58.3 | 81.2 | 29.6 | 63.4 | 82.5 | 38.2 | 57.1 | 84.6 | 42.9 | 60.2 | 87.1 | 44.6 | 62.3 | 87.5 |
| TesNet (Wang et al., 2021) | 32.3 | 66.7 | 82.3 | 38.6 | 67.5 | 84.3 | 40.8 | 64.1 | 84.4 | 51.2 | 66.7 | 86.8 | 55.3 | 68.9 | 88.3 |
| Huang et al. (Huang et al., 2023) | 54.7 | 70.2 | 85.3 | 57.2 | 70.9 | 85.0 | **65.7** | 67.5 | 86.9 | 68.6 | 71.4 | 89.9 | 67.4 | 74.3 | 90.3 |
| *Visual Prompt Tuning Methods* | | | | | | | | | | | | | | | |
| VPT-Shallow (Jia et al., 2022) | 5.6 | 35.5 | 82.4 | 8.9 | 33.6 | 83.5 | 9.0 | 39.2 | 86.2 | 12.7 | 40.3 | 88.5 | 11.3 | 44.5 | 88.7 |
| VPT-Deep (Jia et al., 2022) | 7.7 | 37.1 | 82.7 | 9.2 | 37.8 | 84.0 | 11.5 | 40.6 | 86.5 | 14.6 | 39.5 | 89.1 | 14.0 | 47.6 | 89.5 |
| E$^2$VPT (Han et al., 2023) | 13.7 | 45.2 | 83.8 | 26.7 | 41.9 | 84.3 | 22.3 | 43.2 | 86.7 | 27.5 | 54.3 | 89.3 | 27.3 | 55.0 | 89.4 |
| Gated Prompt Tuning (Yoo et al., 2023) | 8.6 | 32.7 | 84.1 | 16.8 | 28.9 | 84.5 | 28.5 | 41.2 | 86.8 | 34.6 | 60.1 | 89.5 | 35.7 | 61.5 | 89.7 |
| VPT-Shallow (w/ Proto.) | 51.0 | 65.0 | 83.4 | 50.9 | 62.4 | 84.9 | 52.0 | 67.1 | 87.1 | 60.5 | 68.5 | 89.9 | 54.5 | 71.7 | 90.1 |
| VPT-Deep (w/ Proto.) | 54.8 | 66.7 | 84.1 | 59.7 | 70.3 | 85.6 | 61.8 | 65.3 | 87.3 | 70.2 | 72.5 | 90.3 | 61.9 | 75.0 | 90.7 |
| E$^2$VPT (w/ Proto.) | 60.8 | 65.2 | 85.2 | 54.7 | 69.5 | 86.1 | 62.4 | 67.3 | 87.8 | 66.6 | 70.9 | 90.1 | 64.6 | 76.1 | 90.7 |
| Gated Prompt Tuning (w/ Proto.) | 56.7 | 68.2 | 85.2 | 61.3 | 65.4 | 86.2 | 65.4 | 68.2 | 87.8 | 70.3 | 71.8 | 90.4 | 67.2 | 75.6 | 90.5 |
| **IVPT** | **63.1** | **73.4** | **86.2** | **64.5** | **72.3** | **86.7** | 63.5 | **70.2** | **88.1** | **75.3** | **75.9** | **90.8** | **72.6** | **77.4** | **91.1** |

# 4 EXPERIMENTS

## 4.1 EXPERIMENTAL SETUP

We follow visual prompt tuning (Jia et al., 2022) paradigm for classification, supervised by image-level category labels. We employ three DinoV2 (Oquab et al., 2023) variants (ViT-S, ViT-B, ViT-L) with register tokens (Darcet et al., 2023) and two DeiT (Touvron et al., 2021) variants (ViT-S, ViT-B) as backbones, with images resized to $518 \times 518$. We establish the cross-layer prototypes across the last four layers, with the number of concept prototypes $m$ (including background) set to 17, 14, 11, and 8 per layer, and the number of fused prompts $n$ (as well as coarse-grained prompts) set to 4. Balancing ratios $\lambda_{cls}$, $\lambda_{ps}$ and $\lambda_{con}$ are all set to 1. For interpretability, we evaluate scores with 10 concepts or parts (the second-to-last layer), following the number of the evaluation method in (Huang et al., 2023). The concept prototypes at each layer are all initialized randomly with a standard variation of 0.05. More details are provided in the Appendix.

**Datasets.** We evaluate both the classification accuracy and interpretability on the CUB-200-2011 dataset (Wah et al., 2011), which is the only benchmark with part-level annotations required for fine-grained interpretability evaluation. It contains images of 200 bird species, including 5,994 training images and 5,794 testing images. Each image includes keypoint annotations for 15 different bird body parts. We also perform visualizations on the Gleason-2019 (Nir et al., 2018), the Stanford Cars (Krause et al., 2013) and the FGVCAircraft dataset (Maji et al., 2013). Gleason-2019 is a collection of annotated prostate cancer histopathology images about automated Gleason grading of cancer aggressiveness. Stanford Cars is a dataset consisting of 16,185 images of 196 different car models from 10 car manufacturers. FGVC-Aircraft contains 10,200 aircraft images, with 100 images for each of 102 different aircraft variants. Additionally, we conduct quantitative experiments on two other fine-grained datasets: PartImageNet, which provides part-level annotations for 158 classes from ImageNet, and PASCAL-Part, containing detailed part annotations for objects in the PASCAL VOC.

**Evaluation metrics and baselines.** We evaluate the interpretability of the prompts using the consistency and stability scores defined by (Huang et al., 2023). We compare our approach with two categories of methods. First, we compare it with current state-of-the-art part prototype methods, such as ProtoPNet (Chen et al., 2019), ProtoPool (Rymarczyk et al., 2022), TesNet (Wang et al., 2021), and Huang et al. (Huang et al., 2023), which are evaluated based on category-specific prototypes without sharing between categories. We also validate the effectiveness with several visual prompt tuning baselines (Jia et al., 2022; Han et al., 2023; Yoo et al., 2023), along with variants that incorporate our proposed concept-prototype-based prompt learning after the last layer for classification.

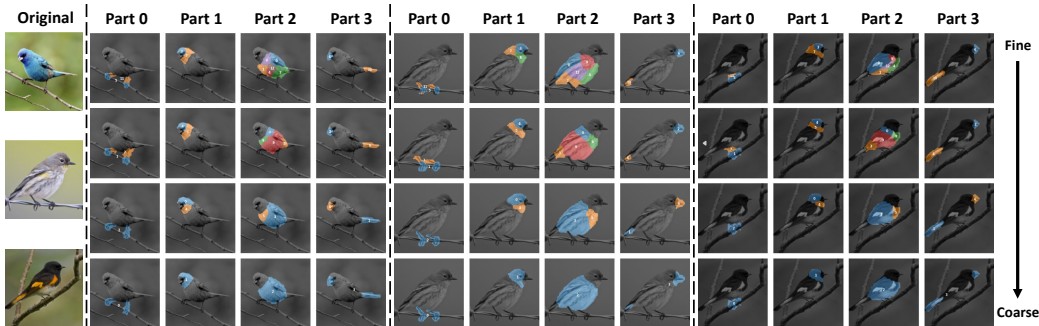

Figure 3: Qualitative results of region maps via the structure of cross-layer prompt fusion.

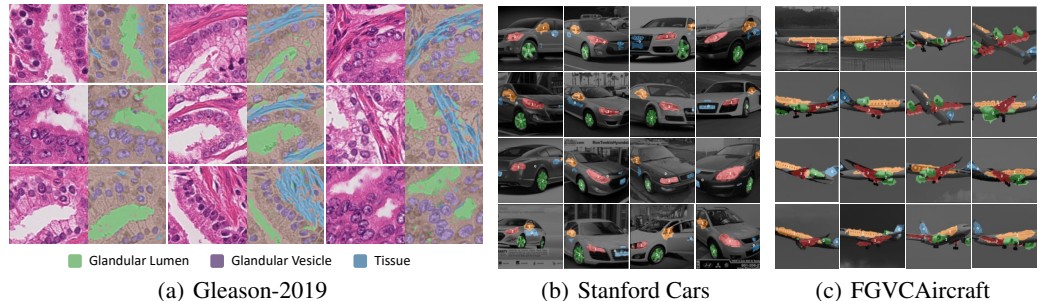

(a) Gleason-2019      (b) Stanford Cars      (c) FGVCAircraft

Figure 4: Qualitative results on patch-level images in the Gleason-2019 dataset about prostate cancer, as well as Stanford Cars and FGVCAircraft about general fine-grained classification, with different colors representing distinct concepts or features relevant to the prediction.

## 4.2 QUANTITATIVE COMPARISONS

We first evaluate the interpretability and accuracy of IVPT with quantitative metrics. Table 1 demonstrates that IVPT outperforms conventional part prototype networks in consistency scores (measuring coherent concept alignment across instances), achieving gains of +8.4%, +7.3%, +6.7%, and +5.2% for DeiT-S, DeiT-B, DinoV2-B, and DinoV2-L, respectively. However, it lags behind Huang et al. (Huang et al., 2023) by 2.2% for DinoV2-S, likely due to smaller models' limited capacity to maintain both interpretability and accuracy. Additionally, IVPT achieves higher stability scores (reflecting robustness to input variations) than prior works, with improvements of 3.2%, 1.4%, 2.7%, 4.5%, and 3.1% for DeiT-S, DeiT-B, DinoV2-S, DinoV2-B, and DinoV2-L, highlighting the framework's resilience to distribution shifts. IVPT also maintains prediction accuracy, even slightly surpassing part prototype networks in recognition performance. Notably, the distinct trends in consistency and stability metrics suggest excellent semantic alignment fidelity and prediction invariance under perturbations.

## 4.3 QUALITATIVE INTERPRETABILITY ANALYSIS

**Qualitative results of the cross-layer structure.** To validate the effectiveness of the cross-layer structure in capturing fine-to-coarse semantics and demonstrate how it addresses the limitations of non-cross-layer concept learning, we provide qualitative visualizations of region maps from three sample images across different layers. As shown in Figure 3, fine-grained semantics at lower layers progressively transition to coarse-grained representations at higher layers, enabling inter-layer interaction and a systematic explanation pathway between detailed and abstract semantics. By leveraging this cross-layer relationship, IVPT efficiently extracts intra-region features to construct interpretable prompts, thereby improving semantic alignment across granularities in complex visual scenes. These results explicitly illustrate how IVPT's cross-layer design enhances its capacity to model and explain multi-granularity concepts.

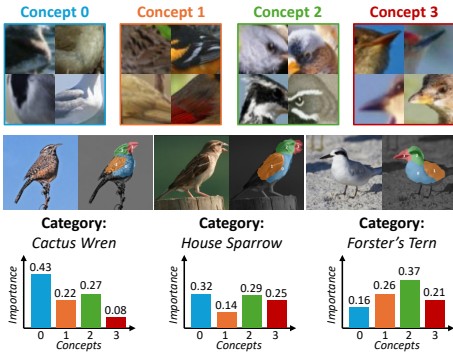

Figure 5: Illustration of coarse-grained region maps with prototypes across images highlighting the importance score of each concepts.

Table 2: Performance comparison on PartImageNet and PASCAL-Part datasets with DinoV2-B.

| Methods | PartImageNet | | | PASCAL-Part | | |
|---|---|---|---|---|---|---|
| | Con. | Sta. | Acc. | Con. | Sta. | Acc. |
| ProtoPool | 47.5% | 56.9% | 61.9% | 52.3% | 45.9% | 75.1% |
| Huang et al. | 58.6% | 62.3% | 66.9% | 67.2% | 68.6% | 78.9% |
| VPT-Deep | 54.2% | 63.7% | 73.9% | 61.5% | 62.9% | 85.9% |
| **IVPT** | **63.2%** | **71.5%** | **74.2%** | **72.6%** | **77.4%** | **86.4%** |

Table 3: Ablation on component combinations.

| Methods | Con. | Sta. | Acc. |
|---|---|---|---|
| Baseline | 62.7 | 64.3 | 88.4 |
| + Spatial bias maps | 63.5 | 66.7 | 88.7 |
| + Intra-region feature aggregation | 65.4 | 68.3 | 89.8 |
| + Cross-layer prototype | 70.4 | 70.9 | 90.5 |
| + Fine-to-coarse prompt fusion | 75.3 | 75.9 | 90.8 |

**Qualitative results on other recognition tasks.** To assess the applicability of IVPT in pathology, we perform explainability experiments on the Gleason-2019 dataset for prostate cancer classification. Whole-slide images are divided into 256×256-pixel patches, each classified with Gleason scores of 3, 4, or 5. As shown in Figure 4(a), IVPT effectively highlights key grading features, such as green regions (glandular lumen) and purple regions (diseased glandular vesicles), consistent with pathological standards. Blue areas denote common tissue types. This visualization underscores IVPT's potential to enhance diagnostic workflows by clarifying the model's reasoning. Further analysis of concept significance across grades is provided in the Appendix.

Moreover, we identify key contributing concepts in two fine-grained recognition datasets: Stanford Cars (Krause et al., 2013) and FGVCAircraft (Maji et al., 2013). As shown in Figure 4(b)-(c), IVPT focuses on fine-grained details essential for distinguishing similar classes. For Stanford Cars, it identifies critical concepts such as the emblem or handle (blue), rearview mirror (orange), wheel (green), and headlight (red). Similarly, for FGVCAircraft, it detects key components including the tail (blue), fuselage (orange), turbine (green), and wing (red). The concepts associated with individual prototypes tend to cluster around semantically related features. When necessary, fine-grained distinctions can be uncovered by subdividing these clusters, enabling interpretable refinement without evidence of uncontrolled polysemantic mixing.

**Importance scores of explainable concept-level regions.** We analyze the importance scores linked to concept-level regions associated with prototypes for classification in Figure 5. The analysis reveals how these four concept prototypes affect classification across three bird species. Each image is associated with a region map, where patches are assigned to one of the regions (concepts), each contributing differently to the classification. Concept-level importance scores are computed via Equation 4 and displayed as bar graphs, showing each concept's varying influence. For instance, Concept 0 has a high score for Cactus Wren (0.43), while Concept 2 is more influential for Forster's Tern (0.37). The region corresponding to the highest-scoring concept overlaps precisely with the discriminative concept of the image, which aids the interpretation of critical areas for classification.

## 4.4 RESULTS ON PARTIMAGENET AND PASCAL-PART

To validate the generalization of our IVPT beyond CUB, we conduct additional quantitative experiments on two fine-grained part-annotation datasets: PartImageNet and PASCAL-Part. Following the evaluation protocol in this paper, a single representative point was extracted for each non-background semantic part (e.g., the centroid of a certain part in PartImageNet). As shown in Table 2, IVPT achieves superior interpretability and robustness on key metrics across datasets, along with a consistent edge in classification accuracy, confirming that our prototype learning framework effectively captures monosemantic concepts in diverse scenarios.

Beyond the quantitative metrics, we present a qualitative analysis to visually demonstrate the concept discovery capability of IVPT. Figure 6 showcases example visualizations from the PartImageNet and PASCAL-Part datasets. Its ability to discover and leverage semantically consistent concepts across

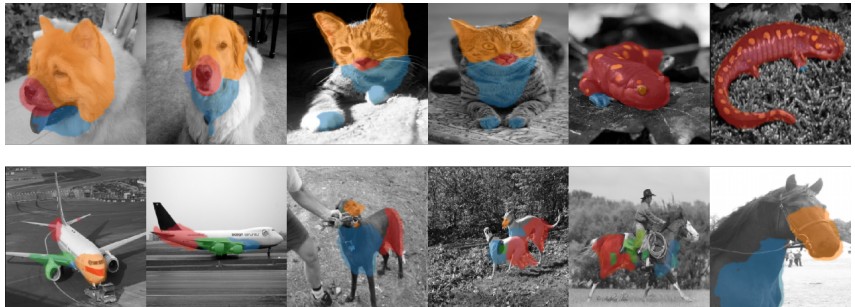

Figure 6: Qualitative analysis on PartImageNet (top) and PASCAL-Part (bottom).

object categories with vastly different morphologies. For instance, as visualized in Figure 6 (bottom), the model trained on PASCAL-Part identifies "head" as a coherent concept for both rigid objects like airplanes and articulated animals like horses and dogs. This demonstrates that IVPT grounds its reasoning in high-level semantics rather than low-level textures or shapes specific to a category. The spatial region maps show that the model precisely localizes these parts (e.g., the nose of a plane, the head of a horse) and associates them with the same underlying concept. This cross-category concept sharing suggests that IVPT builds a rich, part-based representation of the world. During inference, the presence of such universal concepts (e.g., "head", "leg", "body") provide a robust, interpretable evidence trail, validating the generalization of IVPT's interpretability.

### 4.5 ABLATION STUDIES

We first investigate the impact of various components on model performance, as illustrated in Table 3. The baseline applies prompt learning for explainability exclusively in the last layer, using a feature aggregation strategy with global-level attention instead of conditioning on specific regions. Each additional component consistently improves performance across all metrics. Specifically, intra-region feature aggregation achieves a notable improvement in classification accuracy, with an increase of over 1%, as learning features within specific regions enhances their discriminative power. Furthermore, the cross-layer prototypes as well as the fine-to-coarse prompt fusion results in a substantial improvement of both the consistency score and stability score, validating the effectiveness of cross-layer explainability. We also provide ablations on the number of prompted layers and concept prototypes. Please refer to the Appendix for more details.

### 4.6 HUMAN STUDIES

We conduct a comprehensive human evaluation study with 20 participants to validate IVPT's interpretability, demonstrating strong alignment between learned prototypes and human-understandable concepts. The study achieves 97.5% accuracy in concept annotation and high ratings across three key dimensions: detail preservation (4.7/5), semantic abstraction (4.8/5), and transition naturalness (4.8/5), indicating that IVPT's hierarchical concept learning effectively mirrors human cognitive processes. Complete methodological details and illustrative examples are provided in the appendix.

## 5 CONCLUSION

In this paper, we propose Interpretable Visual Prompt Tuning (IVPT) framework, which enhances the interpretability of visual prompt tuning by associating prompts with human-understandable visual concepts through concept prototypes. With a novel cross-layer structure, IVPT aligns concepts across multiple layers for learning explainable prompts. Extensive evaluations demonstrate that IVPT enhances both interpretability and accuracy compared to other methods, highlighting IVPT's potential to enable transparent and effective AI analysis, especially in critical applications. Our limitations include reliance on in-domain concept prototypes, which limit flexibility in diverse domains.

## 6 ACKNOWLEDGEMENT

This work was supported by National Natural Science Fund of China (No. U25A20527, 62473286). This work was also supported by Shanghai Municipal Science and Technology Major Project (No. 2025SHZDZX025G10). This work was supported in part by the National Natural Science Foundation of China under Grants 62576262, in part by the Key Research and Development Program of Shaanxi Province under grant 2024SF-YBXM-647.

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

# A  ADDITIONAL TRAINING DETAILS

We train all our models using the Adam optimizer, with class token, position embedding, register token (in DinoV2) and distillation token (in DeiT) unfrozen. We use a starting learning rate of $2 \times 10^{-4}$ for the fine-tuned tokens of ViT backbone, spatial bias maps and learnable concept prototypes, and $1 \times 10^{-2}$ for the other tunable layers of the model, including the fusion layers for prompt generation and the linear head for classification, with a batch size of 16. Training lasts for a total of 25 epochs with a duration of 2 hours on 4 NVIDIA RTX A6000 GPUs, and we employ a step learning rate schedule, reducing the learning rate by a factor of 0.5 every 4 epochs. The concept prototypes at each layer are all initialized randomly with a standard variation of 0.05.

# B  DETAILS OF PART-SHAPING LOSS

While the classification loss $\mathcal{L}_{cls}$ ensures the discriminative capability of the discovered parts, it does not explicitly guide the attention maps to focus on semantically salient regions of the object. To mitigate this limitation, we follow the methodology outlined in (Aniraj et al., 2024), which proposes a set of additional objective functions that incorporate structural priors into the learning process, summed as the part-shaping loss $\mathcal{L}_{ps}$. Below, we provide a detailed description of each loss function.

## B.1  ORTHOGONALITY LOSS

Orthogonality loss $\mathcal{L}_{\perp}$ encourages decorrelation among the learned part embedding vectors by minimizing their pairwise cosine similarity. Formally, for the modulated embedding vectors $\mathbf{v}_m^k$, the loss is computed as:

$$\mathcal{L}_{\perp} = \sum_{k=1}^{K+1} \sum_{l \neq k} \frac{\mathbf{v}_m^k \cdot \mathbf{v}_m^l}{\|\mathbf{v}_m^k\| \cdot \|\mathbf{v}_m^l\|} \tag{11}$$

## B.2  EQUIVARIANCE LOSS

Equivariance loss $\mathcal{L}_{\text{eq}}$ ensures consistent part detection under geometric transformations. This is achieved by: (1) applying a random affine transformation $T$ to the input image, (2) processing both original and transformed images through the network, (3) inverse-transforming the attention maps from the transformed image, and (4) computing the cosine distance between the original and transformed attention maps. This formulation encourages the learned parts to be equivariant to rigid transformations like translation, rotation, and scaling. This loss is mathematically defined using the attention map function $A^k(\mathbf{x})$, which generates the $k^{th}$ attention map for input image $\mathbf{x}$. The loss computes the normalized correlation between the original attention maps and those from transformed images after inverse transformation:

$$\mathcal{L}_{\text{eq}} = 1 - \frac{1}{K} \sum_k \frac{\left\| A^k(\mathbf{x}) \cdot T^{-1} \left( A^k \left( T(\mathbf{x}) \right) \right) \right\|}{\|A^k(\mathbf{x})\| \cdot \|A^k \left( T(\mathbf{x}) \right)\|} \tag{12}$$

## B.3  PRESENCE LOSS

The presence loss consists of two components designed to enforce different spatial priors:

- Foreground presence loss $\mathcal{L}_{p_1}$ ensures each foreground part appears in at least some images within a mini-batch. For a batch $\{\mathbf{x}_1, ..., \mathbf{x}_B\}$, it operates on pooled attention maps $\bar{A}^k(\mathbf{x}_b)$ to prevent single-pixel solutions:

$$\mathcal{L}_{p_1} = 1 - \frac{1}{K} \sum_k \max_{b,i,j} \bar{a}_{ij}^k(\mathbf{x}_b) \tag{13}$$

- Background presence loss $\mathcal{L}_{p_0}$ enforces consistent background detection across all images, with a spatial bias toward image boundaries through a soft mask $M$:

$$\mathcal{L}_{\text{p}_0} = -\frac{1}{B} \sum_b \log \left( \max_{i,j} \, m_{ij} \bar{a}_{ij}^{K+1}(\mathbf{x}_b) \right) \tag{14}$$

The mask weights $m_{ij}$ increase radially from the image center:

$$m_{ij} = 2 \left( \frac{i-1}{H-1} - \frac{1}{2} \right)^2 + 2 \left( \frac{j-1}{W-1} - \frac{1}{2} \right)^2 \tag{15}$$

### B.4  ENTROPY LOSS

Entropy loss $\mathcal{L}_{\text{ent}}$ encourages unambiguous part assignments by minimizing the entropy of attention distributions:

$$\mathcal{L}_{\text{ent}} = \frac{-1}{K+1} \sum_{k=1}^{K+1} \sum_{ij} a_{ij}^k \log \left( a_{ij}^k \right) \tag{16}$$

### B.5  TOTAL VARIATION LOSS

Total variation loss $\mathcal{L}_{\text{tv}}$ promotes spatial coherence in attention maps without imposing strict shape constraints, favoring connected components through standard total variation regularization. This loss is employed to encourage spatial smoothness in the learned part attention maps. This regularization term computes the average magnitude of spatial gradients across all part maps:

$$\mathcal{L}_{\text{tv}} = \frac{1}{HW} \sum_{k=1}^{K+1} \sum_{ij} |\nabla a_{ij}^k| \tag{17}$$

where $\nabla a_{ij}^k$ represents the spatial gradient (computed via finite differences) at position $(i, j)$ in the $k^{th}$ attention map $A^k$. This formulation promotes the formation of coherent regions in the attention maps while remaining agnostic to specific part shapes. The normalization by image dimensions ensures scale-invariant regularization across different input resolutions.

## C  ANALYSIS ON CRD/IFA COMPUTATIONAL COST DURING TRAINING/INFERENCE

IVPT introduces a moderate computational overhead compared to vanilla Visual Prompt Tuning (VPT), primarily due to the incorporation of its Concept Region Discovery (CRD) and Intra-region Feature Aggregation (IFA) modules. When implemented in PyTorch and executed on an NVIDIA RTX 3090 GPU, the attention mechanism and additional loss computations within CRD result in an approximate 4.8% increase in training time and a 5.2% rise in per-image inference latency. This overhead is largely attributable to the cross-layer prototype similarity calculations, whose complexity scales with the feature dimensions.

Notably, the introduced modules are designed to be lightweight, contributing only a minimal number of tunable parameters to the overall model. The breakdown of parameters is as follows:

- **CRD Tunable Parameters:**
    - Concept Prototypes: 50 in total distributed across layers (i.e., $17 + 14 + 11 + 8$)
    - Each prototype is a 768-dimensional vector
    - $\rightarrow$ Total: $50 \times 768 = 38,400$ parameters
    - Spatial Bias Maps: 50 concept-specific maps at a resolution of $37 \times 37$
    - $\rightarrow$ Total: $50 \times 37 \times 37 = 68,450$ parameters
    - **Total CRD Parameters:** 106,850
- **IFA:** No tunable parameters (utilizes fixed feature averaging)

For context, the full Vision Transformer-Base (ViT-B) backbone contains approximately 86 million trainable parameters (based on the standard DinoV2 ViT-B configuration). The additional parameters introduced by CRD and IFA account for only about 0.12% of the total parameters required for full ViT fine-tuning. Thus, IVPT achieves significant gains in interpretability while maintaining minimal computational overhead.

## D    EXTRA ABLATIONS

**Ablation on the hierarchical structure of prototypes.**    We investigate how the hierarchical prototype structure strategy affects the performance, as shown in Table 4. We use DinoV2-B as the backbone (the same below). When using only one layer, the model exhibits poor interpretability and accuracy. Increasing the number of layers improves performance across all metrics, with four layers achieving optimal results, suggesting enhanced inter-layer relationships that improve interpretability. However, applying explainable prompt learning to more layers introduces semantic confusion and reduces performance, due to fewer explainable concepts in feature maps from shallower layers. Notably, reducing the number of prototypes layer by layer enhances interpretability by capturing concepts at varying granularities, facilitating dynamic prompt modeling.

Table 4: Ablation on the number of prompted layers and the number of prototypes at each layer. Bolded items in each list indicate layers used for interpretability evaluation, each with 10 prototypes. After the final layer, we consistently use 4 prototypes to generate coarse-grained part features for the final classification.

| Layers | Prototype Number | Con. | Sta. | Acc. |
|---|---|---|---|---|
| 1 | [**10**] | 62.5 | 65.2 | 89.5 |
| 2 | [**10**, 7] | 70.7 | 71.8 | 89.8 |
| 2 | [**10**, 10] | 67.2 | 72.6 | 89.6 |
| 3 | [13, **10**, 7] | 73.7 | 73.0 | 90.3 |
| 3 | [10, **10**, 10] | 74.4 | 70.7 | 90.5 |
| 4 | [16, 13, **10**, 7] | 75.3 | **75.9** | **90.8** |
| 4 | [10, 10, **10**, 10] | **75.6** | 71.2 | 90.5 |
| 5 | [19, 16, 13, **10**, 7] | 66.6 | 68.1 | 90.1 |
| 5 | [10, 10, 10, **10**, 10] | 64.3 | 67.4 | 89.7 |

**Analysis on cross-layer concept alignment.**    To rigorously quantify cross-layer concept alignment, we propose a new experiment measuring the Intersection over Union (IoU) between high-level concept regions and aggregated low-level concept regions. Extensive experiments on the CUB-200-2011 validation set reveal exceptional alignment, as shown in Table 5.

Table 5: Intersection over Union (IoU) measurements between different layer combinations

| Layers | IoU |
|---|---|
| Layer 1 (finest granularity) $\rightarrow$ Layer 4 (coarsest) | 98.7% $\pm$ 0.8 |
| Layer 2 $\rightarrow$ Layer 4 | 98.9% $\pm$ 0.6 |
| Layer 3 $\rightarrow$ Layer 4 | 99.2% $\pm$ 0.4 |
| Overall mean | 99.0% $\pm$ 0.6 |

These results demonstrate that our cross-layer fusion mechanism (Eq. 8-9) achieves near-perfect spatial consistency (IoU >98% universally), validating that fine-grained concepts are cohesively grouped under unified high-level semantics, and that the cross-layer prompt fusion with a concept region consistency loss work well.

## E    HUMAN EVALUATION METHODOLOGY AND RESULTS

We conduct a structured three-stage evaluation with 20 participants to assess IVPT's interpretability. In the concept annotation phase, participants freely describe semantic meanings for 100 prototype-

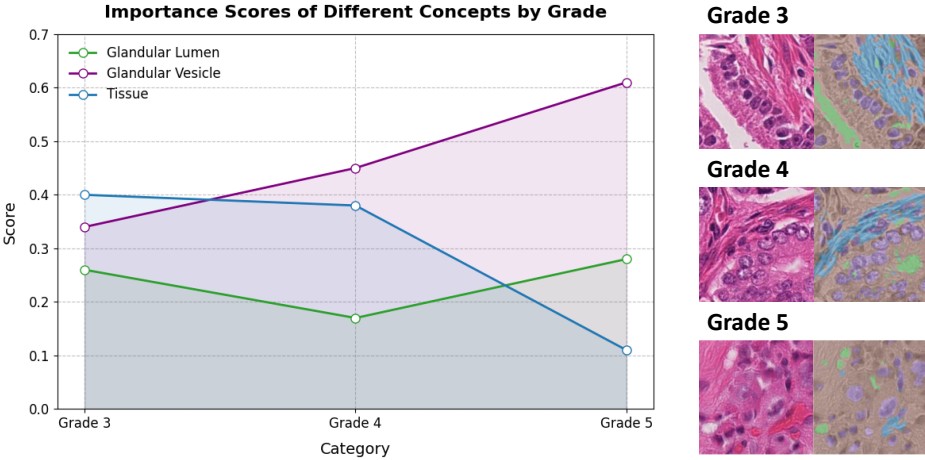

Figure 7: Illustration of importance scores of different concepts by grade on the Gleason-2019 dataset.

activated regions from CUB-200-2011 images, achieving 97.5% overall accuracy in matching human descriptions to prototype intentions.

For hierarchical validation, participants evaluate cross-layer transitions using 5-point Likert scales (1 = strongly disagree, 5 = strongly agree), showing strong consensus across three critical aspects:

**Detail Preservation** (mean = 4.7): Participants confirm IVPT's ability to retain fine-grained visual details during abstraction from shallow to deep layers. Example: When identifying a "Cactus Wren", shallow-layer prototypes capture individual spine-like breast feathers, while deep-layer prototypes preserve these textural details within broader anatomical concepts.

**Semantic Abstraction** (mean = 4.8): Users agree higher layers effectively capture meaningful macro-concepts without distorting original semantics. Example: For a "Black-footed Albatross", shallow prototypes detect detailed "hooked beak tip" features that naturally abstract to coherent "head" concepts in deeper layers.

**Transition Naturalness** (mean = 4.8): The hierarchical progression demonstrates intuitive logical coherence aligned with human reasoning patterns. Example: Observing a "Red-winged Blackbird", users note seamless progression from wing edge curvature (Layer 1) to epaulet shape recognition (Layer 2), mirroring expert birding observation patterns.

The study demonstrates IVPT's effectiveness in bridging the gap between machine representations and human-interpretable visual concepts through its cross-layer prototype architecture.

## F   FURTHER ANALYSIS ON THE GLEASON-2019 DATASET

Figure 7 illustrates the importance scores of different concepts (Glandular Lumen, Glandular Vesicle, Tissue) across grades in the Gleason-2019 dataset. Glandular Vesicle shows a steady increase in importance from Grade 3 to Grade 5, reaching around 0.6, which aligns with the fact that variations in glandular vesicles are a primary indicator of prostate cancer. As the malignancy level increases, its importance grows significantly. In contrast, Glandular Lumen remains consistently low, which is reasonable given its lack of direct relevance to cancer. Meanwhile, Tissue shows a high importance at Grade 3 but decreases significantly by Grade 5, likely due to the differentiation of tissue in highly malignant samples. This pattern suggests that different concepts contribute variably to feature extraction as malignancy progresses.

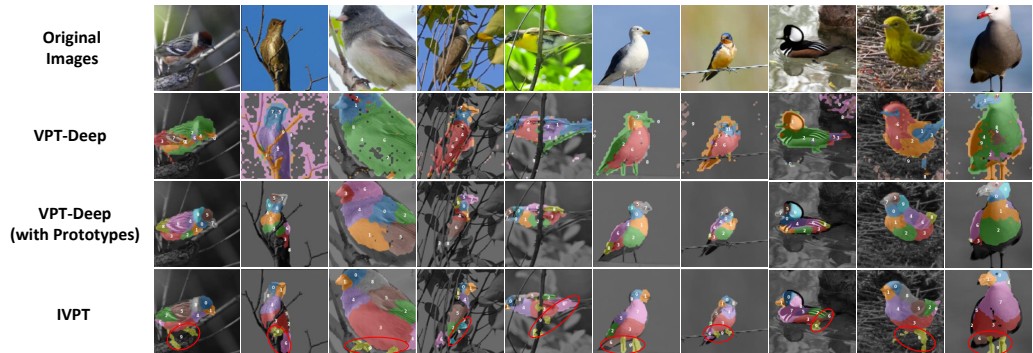

Figure 8: Qualitative results of explainable region maps generated by various approaches. We highlight the areas that significantly distinguish IVPT from other methods with red circles.

## G MORE VISUALIZATION

### G.1 QUALITATIVE RESULTS OF EXPLAINABLE REGION MAPS.

We present qualitative results of explainable region maps generated by different approaches using DinoV2-B as the backbone (the same below) in Figure 8. IVPT outperforms other methods by producing more comprehensive and interpretable maps. Unlike the other two methods with prototype learning, vanilla VPT-Deep struggles to identify explainable regions for the learnable prompts. In comparison to VPT-Deep with prototypes, the red-circled areas in the IVPT row highlight additional essential details, such as finer anatomical features of birds, demonstrating its ability to focus on relevant features for clearer explanations. However, we also notice that in complex scenes with occlusions or dense backgrounds (e.g., CUB images where birds overlap with branches), IVPT prototypes occasionally misalign due to visual ambiguity. For example, we often found that a prototype designed to capture "bird legs" activated on a textured branch mimicking leg scales, incorrectly assigning high importance to background clutter.

### G.2 GENERALIZATION ANALYSIS OF IVPT.

To further assess the generalization of our method beyond fine-grained recognition, we evaluate IVPT on broader image classification benchmarks including CIFAR-100 and ImageNet. As illustrated in Figure 9 (left), IVPT consistently identifies semantically meaningful object parts even in these more generic settings, demonstrating that its part-discovery capability is not restricted to fine-grained domains.

We also explore whether the learned concept prototypes can identify semantically meaningful parts when generalizing on unseen categories (class domain) from the same dataset. As illustrated in Figure 9 (right), when applied to novel categories not encountered during training, IVPT successfully localizes coherent object regions—such as bird heads or wings in fine-grained recognition—without any task-specific retraining. This indicates that the discovered prototypes capture transferable visual concepts that are consistent across category boundaries, rather than overfitting to specific training classes. The results confirm that IVPT exhibits robust cross-category concept generalization, reinforcing the stability and semantic relevance of its learned prototypes.

To investigate the influence of domain shift on the concepts discovered by IVPT, we conduct a comparative visualization study using the VLCS (Fang et al., 2013) benchmark. We fine-tune IVPT separately on two domains—Caltech and PASCAL—using a pre-trained ViT backbone, and visualize the top activated concepts for the "dog" and "person" classes. As shown in Figure 10, these visual comparisons illustrate that IVPT dynamically adapts its concept priorities according to the domain, confirming that the method does not assume a fixed concept set but rather learns domain-relevant interpretations. This behavior makes IVPT suitable for deployment in environments where domain shifts are expected.

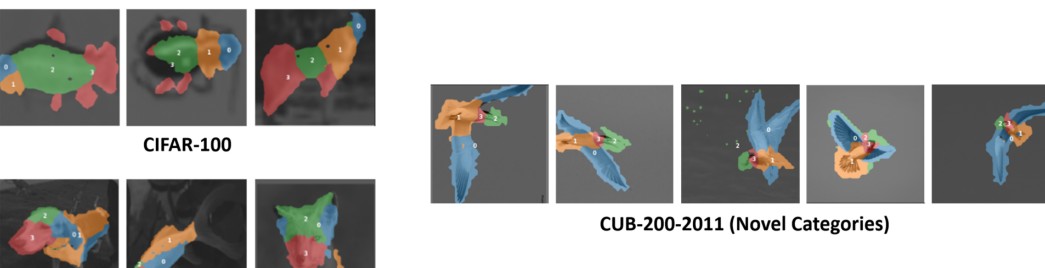

Figure 9: Qualitative results of IVPT on general image recognition datasets (left) and on novel categories from the fine-grained CUB-200-2011 dataset (right).

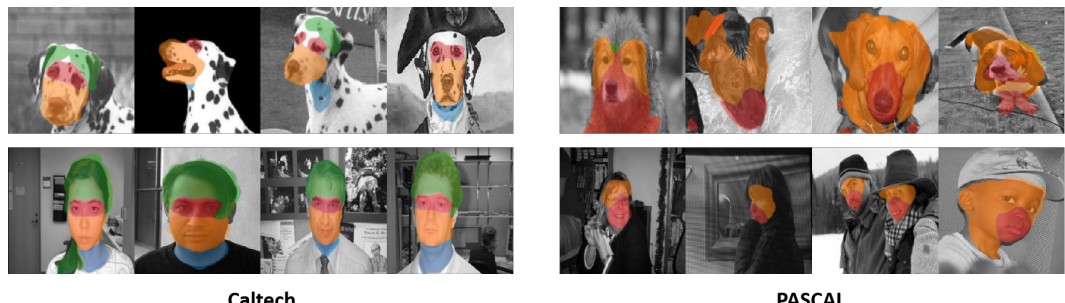

Figure 10: Concept activation visualizations for IVPT models fine-tuned separately on the Caltech and PASCAL domains of the VLCS benchmark. The comparison of highlighted regions shows how the model prioritizes different semantic concepts when the domain shifts.

### G.3 BAD CASE ANALYSIS.

Figure 11 presents representative failure cases of our IVPT model. The analysis of these cases reveals that the model can learn spurious correlations, such as a "bird" prototype activating on surrounding branches or a "shark" concept triggered by water texture. Crucially, this very interpretability not only makes these biases transparent and diagnosable but also directly enables practical model debugging and bias detection. By visually pinpointing and diagnosing such learned biases—where the model relies on context rather than object features, IVPT allows researchers to move from observation to action—facilitating targeted mitigation through data refinement or prototype pruning, thereby solidifying its role in building more robust and trustworthy models.

### G.4 EXPLAINABLE REGION MAPS WITH DIFFERENT NUMBER OF CONCEPT PROTOTYPES.

This section presents additional visualizations to highlight the explainability of region maps with different numbers of concept prototypes. Specifically, Figure 12 shows explainable region maps with varying concept prototypes. As the number of concept prototypes decreases, the region maps simplify, grouping larger areas into fewer explainable regions. These visualizations demonstrate how reducing the number of concept prototypes affects the granularity of the explainable regions, with higher values

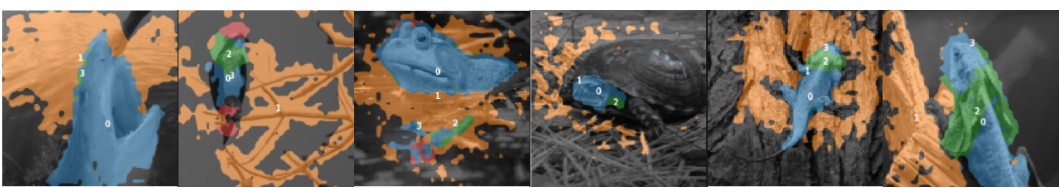

Figure 11: Analysis of IVPT failure cases.

providing finer, more detailed part regions across the bird images. This enables an analysis of how concept-prototype-based explanations adapt to different prototype settings, illustrating the trade-off between granularity and interpretability.

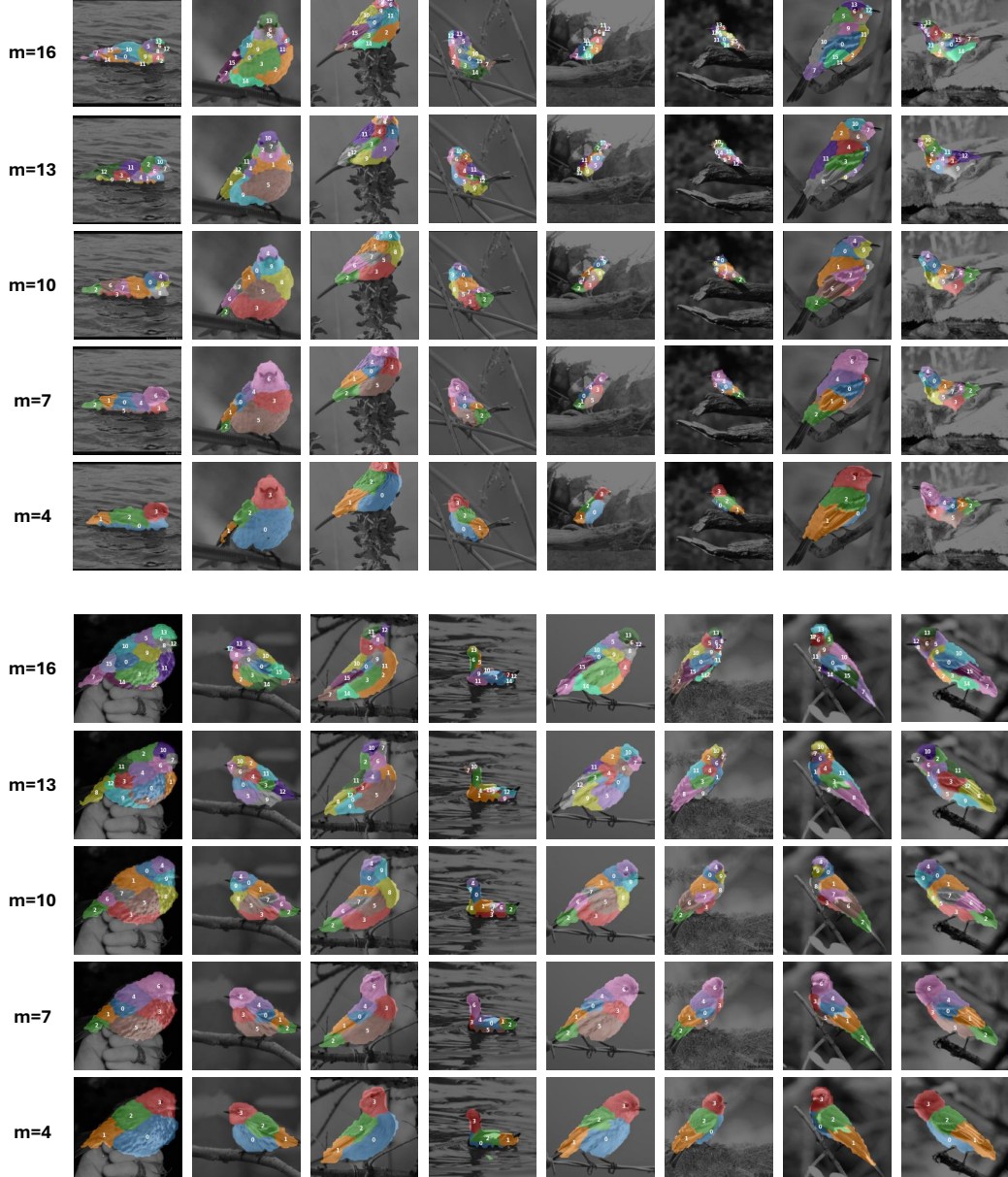

Figure 12: Visualization of explainable region maps with 16, 13, 10, 7 and 4 concept prototypes.

## G.5 EXPLAINABLE REGION MAPS OF THE HIERARCHICAL STRUCTURE USING DIFFERENT BACKBONES.

This section provides the visualization of explainable region maps generated from different backbones with the hierarchical structure, as illustrated in Figure 13. The used backbones include DeiT-S, DeiT-B, DinoV2-S, DinoV2-B, and DinoV2-L. Each row displays how each model processes the image of a bird, capturing distinct regions with different color-coded parts, each corresponding to a concept prototype. The visualization reveals variations in part regions depending on the backbone architecture, with some models offering finer ones than others. These differences highlight the impact of the

backbone choice on the interpretability and granularity of the explainable regions, emphasizing how different architectures contribute uniquely to the generation of part-level explanations in the hierarchical structure. This comparative visualization demonstrates the flexibility and adaptability of the explainable region maps across varying backbones.

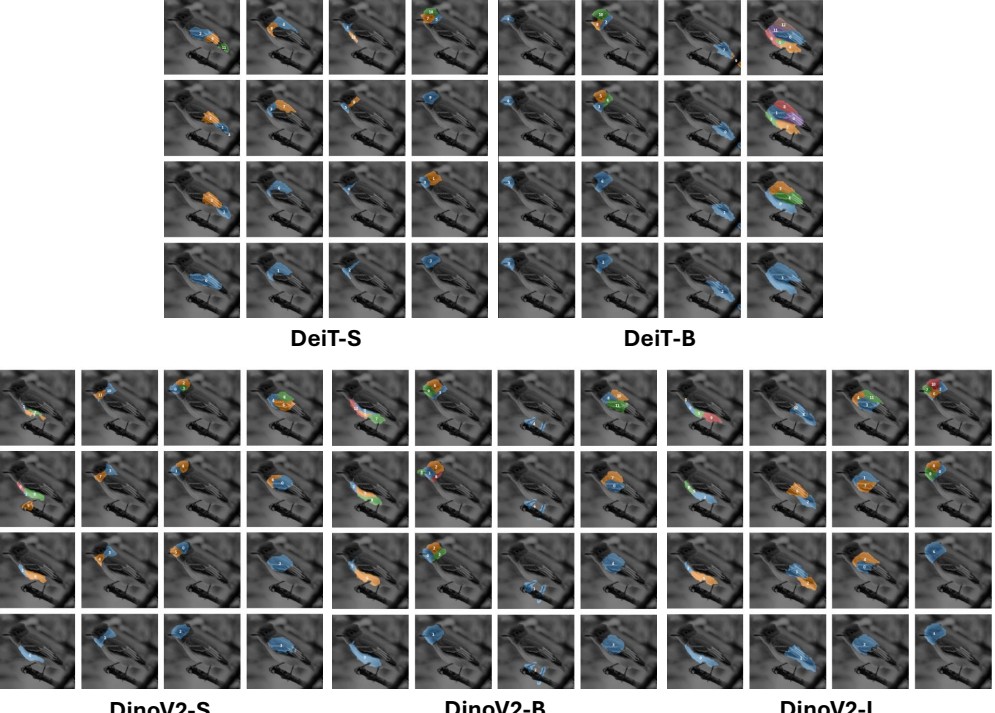

Figure 13: Visualization of explainable region maps of the hierarchical structure using different backbones, including DeiT-S, DeiT-B, DinoV2-S, DinoV2-B, and DinoV2-L.

