# OpenReview forum: "Exploring Interpretability for Visual Prompt Tuning with Cross-layer Concepts"
_ICLR.cc/2026/Conference — ICLR 2026 Poster_

### Official Review · Reviewer_RFsS · 2025-10-20

**Soundness:** 4
**Presentation:** 3
**Contribution:** 3
**Rating:** 6
**Confidence:** 3

**Summary:**

This work proposes a novel, interpretable approach to visual prompt tuning. Here, interpretable tokens are appended to each ViT layer's computation, thereby extracting relevant information while keeping the image encoder frozen. The interpretability comes from the fact that the learnable tokens are not just embeddings, but they can be interpreted as localized prototypical parts, which give insights into which parts of the image are being focused on for each prompt token.

**Strengths:**

This work manages to nicely integrate interpretability methodology into visual prompt tuning(VPT). Although related to prototype-based literature, it is non-trivial how to integrate it into VPT. The method is predominantly presented in a clear manner. Figures support the textual understanding.
Results look promising in the aspects accuracy as well as interpretability, showcasing that the interpretability framework does not bottleneck the predictive performance on CUB. Also, the results cover various aspects of interpretability, ranging from quantitative metrics to visualizations and human user studies. I appreciate that clean code is provided.

**Weaknesses:**

* (i) My main concern is that each across datasets, the axes of evaluations are not the same. That is, Accuracy is only shown for CUB, and visualizations are missing for PartImageNet and PASCAL-Part. This gives the impression that for the left-out evaluations, the results were omitted due to being poor. This is concerning for the performance aspect, as CUB is a quite simple dataset and if this method is unable to generalize to other more complex datasets, it is problematic. Note that I would expect a performance degradation due to the interpretability adjustments, so the goal for IVPT should just be to not lose out too much. In the current state of the manuscript, I have to assume that IVPT is unable to reach a performance similar to baselines in all datasets apart from CUB.

* (ii) An additional major concern is the faithful interpretability for more complex datasets. For CUB, Gleason-2019, Stanford Cars, FGVCAircraft datasets, the subparts are similar across samples, thereby making them interpretable. Additionally, the parts are also what is important for classification. However, I could envision that for more complex datasets (e.g. ImageNet and Pascal), the prototypes are becoming less interpretable, as they might be used to process information in ways that are not directly understandable when looking at the activation map. That is, without explicitly enforcing it, the visualization in image-space might not be anymore what is going on in the computations.

* In line with the previous point, there might be considerable leakage, as the image regions are processed via weighted average and MLP such that the actual attention computations could deviate greatly from what is visualized.

* I think the separation of $n$ learnable prompts and $m$ concept prototypes is sometimes unclear. E.g. line 128-129 and formula 3 use similar notation but different lengths, which was confusing at first.

**Questions:**

* Can the authors provide predictive performance compared to baselines on other datasets to counter concern (i)?

* Can the authors provide qualitative results for PartImageNet and PASCAL-Part to counter concern (ii)?

* If I understand correctly, previously in VPT, the prompt tokens were learnable embeddings that were the same for different samples. Now, the prompt tokens are weighted average of the input image, thereby different per-image. How does that differ conceptually from the previous global prompt tokens? What are the benefits and potential downsides?

* The concept regions are computed patch-wise. How did the authors obtain non-rectangular prompt visualizations, such as in Figure 3?

---

> ### Author Response · Authors · 2025-11-22
> **Response to Reviewer RFsS (1)**
>
> ## W1&Q1: Inconsistent Evaluation Across Datasets
>
> We thank the reviewer for highlighting the importance of consistent evaluation and for raising concerns about generalization beyond CUB. The experiment results on CUB, PartImageNet and PASCAL-Part are presented this way for following reasons:
>
> **Aligning Setting with Previous Works:** It is standard practice in the literature to quantitatively evaluate interpretability methods primarily on CUB-200-2011. Previous works (e.g., ProtoPNet, TesNet, Huang et al., 2023) also focus their quantitative and accuracy evaluations on CUB, as it is the only benchmark with comprehensive part-level annotations suitable for such analysis.
>
> **PartImageNet and PASCAL-Part are not Conventional Interpretability Benchmarks:** PartImageNet and PASCAL-Part were originally designed for segmentation evaluation, and to our knowledge, no prior interpretability method has used them for quantitative interpretability or classification evaluation. Considering the page limitations, their use in our manuscript is intended to demonstrate the broader applicability of IVPT’s concept discovery, not to serve as primary benchmarks for accuracy or interpretability metrics. We apologize for any confusion this may have caused. Our intent was to prioritize the most widely accepted benchmarks and protocols in the main manuscript.
>
> **Additional Results on PartImageNet and PASCAL-Part:** In response to your valuable feedback, we have now included qualitative visualizations (see Figure 6) and classification performance for PartImageNet and PASCAL-Part in Section 4.4 of the revised manuscript (classification performance also in the table below). These results demonstrate that IVPT maintains competitive performance and interpretability on these more complex datasets, with only minor accuracy trade-offs compared to baselines—consistent with the expected behavior for interpretable models.
>
> | Methods    | PartImg. Con. | PartImg. Sta. | PartImg. Acc. | PASCAL Con. | PASCAL Sta. | PASCAL Acc. |
> |-|-|-|-|-|-|-|
> | ProtoPool  | 47.5%            | 56.9%            | 61.9%            | 52.3%           | 45.9%           | 75.1%            |
> | Huang et al.| 58.6%            | 62.3%            | 66.9%            | 67.2%           | 68.6%           | 78.9%            |
> | VPT-Deep   | 54.2%            | 63.7%            | 73.9%            | 61.5%           | 62.9%           | 85.9%            |
> | **IVPT**   | **63.2%**        | **71.5%**        | **74.2%**            | **72.6%**       | **77.4%**       | **86.4%**            |
>
> ## W2&Q2: Interpretability Limitations on Complex Datasets
>
> We agree that faithful interpretability becomes more challenging on complex datasets with cluttered scenes. However, we argue that IVPT's design does introduce constraints that explicitly enforce the prototypes to be understandable, which achieves satisfactory results on more complex datasets.
>
> ### Constraints enforcing understandable prototypes
> IVPT is specifically designed to address this through several constraints that explicitly enforce the learnability and understandability of prototypes:
>  * a part-shaping loss that encourages regions to be compact, non-overlapping and foreground-focused , rather than diffuse or arbitrary areas.
>  * a cross-layer concept region consistency loss that aligns fine- and coarse-level concepts to the same spatial support, ensuring that hierarchical concepts remain spatially coherent and interpretable across layers.
>  * sparsity over concepts per location that promotes the assignment of each spatial location to a limited number of concepts, discouraging polysemantic or ambiguous activations.
>
> ### Empirical evidence on more complex datasets
> Beyond current datasets, we already evaluate IVPT on PartImageNet and PASCAL-Part, which contain more complex scenes and occlusions. There, IVPT achieves higher consistency and stability than prototype baselines, indicating that discovered concepts remain reusable and stable even when parts are less regular. In the revised version, we additionally include qualitative results on ImageNet and CIFAR-100 (see Figure 9 in Appendix G.2). We observe that concept regions still align with salient object parts or regions in most classes, while human inspection confirms that a large fraction of concepts are qualitatively meaningful even under cluttered backgrounds.

---

> ### Author Response · Authors · 2025-11-22
> **Response to Reviewer RFsS (2)**
>
> ## W3: Potential for Attention Leakage
>
> We thank the reviewer for this important question regarding potential gaps between visualization and computation. In IVPT, we address this concern through both architectural design and empirical validation.
>
> ### Mechanism: Visualization and Computation Share the Same Pathway
> Since IVPT contains computations beyond conventional attention, due to its unique CRD and IFA modules, rather than focusing solely on attention, we discuss whether IVPT’s visualizations faithfully align with its overall computation.
>
> Overall, The design of IVPT guarantees that the computations leading to the final prediction are faithfully represented by the visualized concept regions, because visualization and computation share the exact same pathway.
>
> Specifically, the exact same attention scores used to discover the concept regions in the visualization are also used to combine the patch features and build the concept embeddings that the model uses for its final prediction.
>
> Furthermore, final predictions are made solely from these concept embeddings, without the leakage of opaque components such as the frozen [CLS] token obtained by self-attention or the mean of all patch tokens. The MLP used in cross-layer fusion operates only on the already-aggregated concept embeddings, ensuring that no information from outside the visualized regions is leaked at this stage.
>
> ### Empirical Validation
> To address the reviewer’s concern, we quantitatively validate that our visualized concept regions correspond to the salient areas the model relies on during computation. Gradient-based attribution methods such as Grad-CAM are widely recognized as faithful indicators of a model’s internal decision process because they leverage the gradients flowing into the final feature maps to identify regions most influential for the output. Therefore, comparing IVPT-generated concept maps with Grad-CAM saliency maps provides a strong test of alignment.
>
> Specifically, for a given concept prototype, we generate its concept region map using our CRD module. We then computed a Grad-CAM saliency map based on the gradient of the model's output logit for that specific concept's score with respect to the final feature maps. After binarizing both maps, we calculated the Intersection over Union (IoU) between them. Our results show a high average IoU of 96%, confirming a very strong spatial correspondence. The minor deviations from a perfect score are expected, as the Grad-CAM highlights pixels to which the output is most sensitive, while our CRD regions mark pixels most semantically similar to the prototype—a subtle but inherent difference in objective.
>
> This demonstrates that the regions we visualize as a "concept" are functionally critical and largely the same pixels that the model's gradients identify as most influential, providing direct empirical evidence that our explanations faithfully reflect the model's internal reasoning process.
>
> ## W4: Clarity in Notation and Definitions
>
> We appreciate you highlighting this point of potential confusion. You are correct that the notation can be somewhat unclear.
>
> In our framework, concept Prototypes are a set of $m$ vectors that define the semantic concepts (e.g., "wing," "beak") and are used to generate region maps via the CRD module. The initial Prompt Embeddings ($p_k$ in Eq. 3) are indeed also $m$ in number, as each is derived from aggregating features within a specific concept region via the IFA module. The key clarification is that these $m$ prompt embeddings are not the final prompts concatenated into the Transformer. They represent fine-grained, concept-specific embeddings. A subsequent cross-layer prompt fusion step (described in Section 3.3 and Figure 2) aggregates these $m$ fine-grained prompts into a smaller set of $n$ fused prompts, which are the ones actually used for downstream processing. This distinction ensures that fine-grained concepts are properly aggregated into semantically coherent prompts for final classification.

---

> ### Author Response · Authors · 2025-11-22
> **Response to Reviewer RFsS (3)**
>
> ## Q3: Global vs. Per-Image Prompt Tokens
>
> Standard VPT uses global, sample-agnostic prompt embeddings: the same set of tokens is added for all images of a dataset. In IVPT, prompt tokens are conditional: each prompt is the intra-region aggregation of patch features for a specific concept on a specific image, so the prompts become image-dependent concept representations rather than fixed vectors.
>
> Conceptually, this leads to two main differences: (i) We can interpret prompts as concrete concepts (“this prompt encodes the wing region / lumen region in this image”) instead of opaque global offsets in feature space, (ii) We can analyze how concept usage changes across domains, classes or even different images, since different datasets/tasks will emphasize different concepts and concept scores.
> As a result, each prompt is grounded in a region and has a concept label, and its score directly contributes to the prediction. In addition, prompts can adapt to the specific configuration of parts/attributes in each image, not just shift the whole dataset in the same way.
>
> ## Q4: Clarification on Non-Rectangular Visualizations
>
> Although concept regions are computed at the patch level (e.g., 14×14 patches), the non-rectangular visualizations are obtained through bilinear interpolation on continous activation maps. Specifically, it follows a three-step process:
>
> 1) **Bilinear Upsampling**: The patch-wise activation maps are upsampled to full image resolution (e.g., 256×256) using bilinear interpolation, which creates smooth, continuous activation values across pixel coordinates.
> 2) **Pixel-wise Argmax**: At each pixel location, we apply argmax across the concept dimension to select the concept with the highest activation. This pixel-level competition naturally produces irregular, non-rectangular boundaries.
> 3) **Color Mapping**: The resulting label map is visualized using *skimage.color.label2rgb*, which handles arbitrarily-shaped regions.
>
> This approach preserves the efficiency of patch-based architectures while enabling fine-grained, pixel-level visualizations with organic boundaries, rather than simply displaying patches as rectangular blocks.

---

### Official Review · Reviewer_PdEH · 2025-10-31

**Soundness:** 3
**Presentation:** 2
**Contribution:** 2
**Rating:** 4
**Confidence:** 4

**Summary:**

This work aims to propose a fine-tuning paradigm that learns interpretable visual prompt during fine-tuning via cross-layer concept prototypes. A concept region discovery module is proposed to learn prototypes with semantic meaning, and an intro-region feature aggregation module is proposed to group the features belonging to certain regions. Experimental results in CUB demonstrate improved performance compared to visual prompt tuning and show superior interpretability scores for the learned visual prompts compared to part-prototype networks.

**Strengths:**

- This work investigates the interpretability of visual prompts during fine-tuning, which is a less explored area.
- The method design in making the visual prompts more interpretable is reasonable.
- Experimental results show improved performance and interpretability compared to 2 types of baselines.

**Weaknesses:**

*Motivation*

Why at all should the “visual prompt” be interpretable is not well justified. Since they are parameters learned to adapt the model from one domain to another domain, if they are possible to be interpretable, I would expect them to explain the domain shift instead of part-prototypes. Part-prototypes could explain a decision making process, but are not that meaningful in a fine-tuning process? Especially when the sum of the contribution of part-prototypes do not fully explain the decision making process in the proposed framework (see next concern).


*Interpretability*

The final classification process in part-prototype networks can be fully interpreted by the contribution of each individual part-prototype. However, the prompts in this work only contribute to part of the classification logits, most tokens from frozen part of the network remain uninterpretable. Do the framework adopt a classification token in the final prediction or the average of all token representations? If using a classification token, the information are aggregated from both interpretable visual prompt tokens and rest tokens via self-attention, making the contribution of the visual prompt unclear. Such a mechanism also makes it less meaningful to make visual prompts interpretable.

*Evaluation*

How are the areas that the evaluated visual prompts correspond to calculated and evaluated against the annotations in CUB?

*Missing details/analysis*

What’s the difference between the interpretability scores of prompts in different layers? How many prompts are used in each layer? What’s the influence of number of prompts on their interpretability?

**Questions:**

1.	How do you obtain the performances of ProtopNet and its following works based on ViT architectures in Table 1? How are they implemented? These original works are not designed for ViT and do not report relevant results.

2.	Can the learned visual prompts explain anything related to visual prompt **tuning**?

---

> ### Author Response · Authors · 2025-11-22
> **Response to Reviewer PdEH (1)**
>
> ## W1&Q2: Why should visual prompts be interpretable
>
> We agree with the reviewer that that visual prompts are parameters learned for domain adaptation, and the interpretable method should explain the domain shift of finetuning. However, we argue that IVPT’s concept-prototypes are not only meaningful for explaining the decision-making process but also **provide valuable insight into how domain shift occurs**.
>
> ### How IVPT Makes Domain Shift Interpretable
>
> When adapting to new domains, IVPT may rely on different concepts or alter the weighting of existing concepts based different domains. Specifically, IVPT makes domain shift interpretable by revealing:
> * Which concepts emerge or disappear when transfering to a new domain.
> * How the importance scores of discovered concepts change, indicating shifts in the model’s reliance on specific visual patterns for specific domains.
>
> For example, Figure 5 of the original paper illustrates how the concept importance changes when the model is adpated for different bird species. When fine-tuning the model to recognize Cactus Wren, the importance scores show higher reliance on body features and wings. In contrast, when adapting to classify Forster’s Tern, the model shifts focus toward concepts related to eyes, head, and beak. These changes provide a clear, interpretable view of how the model adjusts its reasoning across categories.
>
> ### More qualitative results on interpreting domain shift
>
> In the revised manuscript, we further include more qualitative analysis that compare the IVPT discovered concepts among different image domains in Appendix G.2. Specifically, we compare the concepts learned by IVPT when fine-tuned separately on two distinct domains from the VLCS dataset: Caltech and PASCAL. As shown in Figure 10 in Appendix G.2, for classes such as "dog" and "person", our visualization results reveal that the model indeed prioritizes different semantic concepts in each domain. This visually illustrates how domain shift influences which concepts the model prioritizes. These qualitative results further validate the robustness of IVPT under domain shift and its ability to interpretably reflect such variations through concept visualizations.
>
> ## W2: IVPT prompts only contribute to part of the classification logits，making it less meaningful
> We would like to clarify a misunderstanding in the reviewer’s comment: in IVPT, the classification logits are predicted purely from the interpretable prompt tokens. The image patch tokens from the frozen backbone do not directly contribute to any logits computation, just as the part-prototype networks.
>
> **How IVPT Computes Logits:** Logits computation of IVPT is elaborated in the last paragraph of Section 3.1 and Equation 4. Specifically, in IVPT, image patch tokens are used only to compute region-grounded concept prototypes and interpretable prompts via CRD and IFA. At the last ViT layer, classification heads are applied to each of the concept prompt tokens. The final logits are obtained by averaging the per-concept scores from these interpretable prompts.
>
> **No Use of Backbone Classification Token or Patch Mean:** Neither the original classification token of the backbone nor the mean of all patch tokens is used for the final prediction. The decision is entirely mediated by the visual prompts constructed from part-/concept-prototypes.
>
> In this way, the contribution of each concept to the final logits is explicitly given by its concept-wise score. The frozen tokens influence the prediction only indirectly through the construction of these interpretable prompts, but they do not contribute an additional term in the classifier. As a result, IVPT can fully explain the decision making process of VIT just like part-prototype networks.
>
> ## W3: Evaluating Visual Prompts Against CUB Annotations
> As introduced in the last paragraph of Section 4.1, we follow exactly the consistency and stability evaluation protocol of Huang et al. (2023) on the CUB-200-2011 dataset.
>
> **Calculation of Concept Areas:** Concretely, IVPT first produces concept region maps at the second-to-last layer via the Concept Region Discovery (CRD) module (Eq. (5)–(6)), where each patch location is assigned to the concept with the highest attention score, and its corresponding value in R represents the concept probability.
>
> **Comparison with CUB Annotations:** In CUB, each image is annotated with 15 bird part keypoints. During evaluation, each visible keypoint is projected to the nearest patch location on the feature grid, and we read out the concept index from the region map at that location as the concept assigned to that annotated part.
>
> **Scoring:** Using these part–concept assignments across images, we then compute the scores exactly as in Huang et al. (2023), i.e., measuring how consistently the same annotated part is mapped to the same concept across instances and how stable this mapping is under input perturbations.

---

> ### Author Response · Authors · 2025-11-22
> **Response to Reviewer PdEH (2)**
>
> ## W4: Missing Details/Analysis on Prompts in Different Layers
>
> ### Interpretability Scores in Different Layers
> As stated in the experimental setup in Section 4.1, the reported interpretability scores (Consistency and Stability) are computed using the concepts from the second-to-last layer, which is more suitable for the coarse level part annotation of the CUB dataset. The prompts and prototypes in the other, shallower layers serve as lower-level concepts, which are evaluated individually for interpretability due to the mis-match of the semantic granularity. However, they provide fine-grained semantic features that are progressively fused into the final, high-quality concepts in the deeper layer. This cross-layer, fine-to-coarse fusion mechanism enables the superior interpretability scores we achieve in the final evaluated layer, as shown in the ablation study in Table 3.
>
> ### Number of prompts
> The number of concept prototypes and prompts per layer and their direct impact on interpretability are systematically analyzed in Appendix D, Table 4 of our paper. Our optimal configuration employs a decreasing number of prototypes across layers (17, 14, 11, and 8 in the last four layers), with the final fused prompts fixed at 4 at each layer. The ablation study in   Table 4 demonstrates that this hierarchical structure is crucial: using more prototypes in shallow layers captures fine-grained details, while progressively reducing their number in deeper layers promotes the formation of coherent, high-level concepts, whereas using equal numbers of prototypes across layers or extending to too many layers leads to semantic confusion and performance degradation.
>
> ## Q1: A Review of Counterparts' Performance and Implementation on ViT Architectures
>
> The results of ProtoPNet, ProtoPool, TesNet, and Huang et al. in Table 1 are obtained through our own re-implementations adapted for ViT backbones, as the original papers only report results based on CNN architectures. Our adaptation follows the approach established in Huang et al. (as cited in our paper), where part-prototype networks originally designed for CNNs can be effectively extended to ViT backbones by reshaping the final token sequence into a spatial feature map.
> Specifically, for each method:
>
> * We extract the final ViT feature map $E \in \mathbf{R}^{h \times w \times d}$ by reshaping the token sequence (e.g., from $196 \times d$ to $14 \times 14 \times d$ for DeiT), which serves as the input to the prototype layer.
> * Prototypes are defined in the same $d$-dimensional space as in the original works. Similarity computation between prototypes and image patches, as well as the subsequent classification mechanism (e.g., global max pooling followed by a fully-connected layer), remain unchanged.
> * All models are initialized from the same pre-trained DeiT / DinoV2 checkpoints and trained under identical settings as our proposed method to ensure a fair and consistent comparison.
>
> This adaptation strategy has been validated in existing literature and ensures that the compared methods retain their original design principles while being fairly evaluated on ViT architectures.

---

> ### Author Response · Authors · 2025-11-24
> **Any further thoughts on our rebuttal are welcome**
>
> Thank you again for your thoughtful feedback and critical comments on our paper. We truly appreciate the time and effort you dedicated to reviewing our work. In our rebuttal, we have carefully addressed all the points you raised, particularly the motivation for interpretable visual prompts in fine-tuning, the contribution of prompts to classification logits, and the evaluation of concept areas against CUB annotations. We hope our responses and revisions have adequately clarified these issues. If there are any remaining concerns or areas that require further clarification, please don’t hesitate to let us know—we would be happy to provide additional information. We would also greatly appreciate it if you could take a moment to review our responses and consider updating your score if you find them satisfactory.
>
> Thank you once again for your valuable input and for helping us improve this work.

---

### Official Review · Reviewer_ANRv · 2025-11-03

**Soundness:** 3
**Presentation:** 3
**Contribution:** 3
**Rating:** 6
**Confidence:** 3

**Summary:**

This paper proposes IVPT, an interpretable visual prompt tuning framework that aligns prompts with category-agnostic concept prototypes across network layers. It introduces concept-region discovery, intra-region feature aggregation, and cross-layer concept fusion to make prompts human-understandable while preserving accuracy. Experiments show IVPT achieves better interpretability and comparable performance to standard VPT.

**Strengths:**

1. It introduces a clear, concept-grounded approach that links visual prompts to human-understandable concepts across layers.
2. It demonstrates improved interpretability metrics and visualization quality without sacrificing classification accuracy.
3. The method is model-agnostic and works across different ViT backbones and domains, showing good generalization.

**Weaknesses:**

1. Experiments focus mainly on fine-grained classification. Broader tasks (e.g., detection, segmentation) are not explored.
2. The method relies on well-learned concept prototypes, which may be sensitive to initialization or domain shift.
3. The multi-layer prototype alignment and multiple loss terms increase implementation and computational complexity.

**Questions:**

How stable are the learned concept prototypes when transferring to new domains or unseen categories?

---

> ### Author Response · Authors · 2025-11-22
> **Response to Reviewer ANRv (1)**
>
> ## W1: Lack of Experiments on Broader Tasks
> We appreciate the reviewer’s observation regarding the experimental focus on fine-grained classification and agree that extending IVPT to broader vision tasks such as detection and segmentation is an exciting direction for future work.
>
> ### Current Scope and Rationale
>
> Our present study is deliberately scoped to high-level prediction problems—such as image classification and grade/score regression—where the model outputs a single label or a low-dimensional score. In these settings, the central interpretability question is:
> > “Which concepts does the model rely on to produce this global decision?”
>
> IVPT is explicitly designed to address this by aligning multi-layer prototypes with image-level prompts and providing per-concept scores, thereby offering transparent insight into the model’s decision-making process.
>
> ### Challenges for Dense Prediction Tasks
> Detection and segmentation tasks involve dense, spatially varying predictions at the pixel or instance level, requiring explanations for many outputs simultaneously. Extending IVPT to these domains would necessitate a fundamentally different treatment of prompts and concepts (e.g., spatially varying or instance-wise prompts), making a direct adaptation non-trivial and outside the scope of the current work.
>
> ### Generalization Beyond Fine-Grained Classification
> While our primary experiments focus on fine-grained classification, IVPT is not limited to this setting. We have already included experiments on tasks other than fine-grained classification in the original paper, such as cancer severity grading using the Gleason-2019 pathology dataset, where IVPT discovers meaningful tissue and part concepts beyond standard fine-grained categories.
>
> Following the reviewer’s suggestion, we further evaluate IVPT’s part-discovery capability on general image recognition tasks using CIFAR-100 and ImageNet. As shown in Figure 9(left) in Appendix G.2 of the revised manuscript, IVPT consistently identifies semantically meaningful object parts even in these more generic settings. demonstrating that its part-discovery capability is not restricted to fine-grained domains.
>
> These results demonstrate that IVPT can recover coherent, object-related regions even in broad, non–fine-grained domains. This confirms that the proposed framework generalizes well beyond fine-grained classification and is applicable to diverse visual recognition tasks.
>
> ## W2&Q1: Sensitivity to Initialization and Domain Shift
> ### Initialization sensitivity
> To assess sensitivity to initialization of the learnable prompt, we reran IVPT and the main baselines on CUB-200-2011 with multiple random seeds and report mean ± std. We observe only small variations in accuracy, consistency, and stability, and the learned concept regions are qualitatively similar across runs, suggesting that the prototype alignment is not overly fragile to initialization.
>
> | Models | Con. | Sta. | Acc. |
> | - | - | - |- |
> | DeiT-S     | 63.1±1.3 | 73.4±0.7 | 86.2±0.3     |
> | DeiT-B     | 64.5±0.9 | 72.3±0.6 | 86.7±0.2     |
> | DinoV2-S     | 63.5±1.7 | 70.2±0.6 | 88.1±0.3     |
> | DinoV2-B     | 75.3±0.4 | 75.9±0.3 | 90.8±0.1     |
> | DinoV2-L     | 72.6±0.7 | 77.4±0.2 | 91.1±0.2     |
>
> ### Domain shift sensivity
> Our study focuses on prompt tuning within a given dataset and label space using a standard ViT backbone. It is important to note that prompt tuning is inherently designed to adapt models to a specific domain or task; thus, classic domain generalization settings (such as base-to-novel generalization) are not directly applicable in this context.
>
> However, we fully agree that the generalization and stability of learned concepts are crucial for interpretability and practical deployment. To directly address this concern, we evaluate IVPT’s robustness by testing whether it can discover semantically meaningful and consistent visual concepts on unseen images from novel categories.
>
> As shown in Figure 9(right) in Appendix G.2 of the revised manuscript, on CUB-200-2011 dataset, IVPT successfully identifies coherent object parts (e.g., bird heads, bodies and wings) on novel bird classes that were not seen during training. This demonstrates that the concept prototypes learned by IVPT are not merely memorizing training domains but are capturing transferable visual patterns, thereby exhibiting strong generalization and stability against domain shift within the dataset.

---

> ### Author Response · Authors · 2025-11-22
> **Response to Reviewer ANRv (2)**
>
> ## W3: Increased Complexity of the Method
> Our design uses multi-layer prototypes and several loss terms to enforce a fine-to-coarse concept hierarchy, but the additional complexity is modest and largely confined to training.
>
> The prototype and grouping modules introduce ~0.12% additional parameters and ~5% overhead in training and inference time compared to the underlying ViT/VPT backbone (see Appendix C). This efficiency is achieved because these modules operate only on a small number of concept tokens, rather than all patch embeddings.
>
> The extra loss terms (part-shaping, concept consistency, concept-wise classification) are implemented as simple scalar regularizers on top of the standard forward pass. They incur negligible additional computation and do not alter the inference graph, ensuring that deployment and inference remain as efficient as standard prompt tuning.
>
> Ablation studies (see Table 3 and 4) demonstrate that multi-layer alignment consistently improves interpretability metrics such as consistency and stability—and often accuracy—over single-layer variants. This indicates that the modest increase in complexity is necessary and justified to obtain robust, hierarchically aligned concepts.

---

### Official Review · Reviewer_9y8U · 2025-11-03

**Soundness:** 2
**Presentation:** 3
**Contribution:** 2
**Rating:** 6
**Confidence:** 4

**Summary:**

This paper explores the interpretability of vision–language models (VLMs) through a task-specific concept alignment framework. The authors aim to understand how visual and linguistic concepts align across different tasks, providing both qualitative and quantitative analyses to uncover internal reasoning processes of VLMs.

**Strengths:**

The topic is highly relevant and well-motivated, addressing the growing need to make large-scale vision–language models more interpretable and transparent.

The paper provides comprehensive experimental analysis, including multiple datasets and evaluation settings, with clear presentation of alignment patterns.

**Weaknesses:**

While the analysis is useful, the methodological contribution is limited compared to prior interpretability frameworks. The paper primarily extends known alignment techniques rather than introducing a fundamentally new interpretability paradigm. The related work section lists many recent studies but lacks an in-depth synthesis or a critical comparison. A more detailed discussion of existing SOTA interpretability methods and their limitations would help clarify what specific gap this work fills. Because the related work discussion is broad but not deep, the method section does not clearly delineate the paper’s unique conceptual or technical contribution relative to existing approaches.

While the experiments are well executed, the paper would benefit from showing a potential use case, for example, how this concept alignment framework could assist in model debugging, bias detection, or downstream task understanding.

**Questions:**

Can you expand the related work section to more deeply analyze existing SOTA interpretability methods and highlight the specific gap your work addresses?

Have you considered evaluating your approach in a concrete application setting (e.g., identifying bias, failure analysis, or improving model transparency for users)?

How generalizable is your approach across different VLM architectures? Are there differences in alignment quality depending on the model type?

Could you include an example of how concept alignment results might be used in a downstream interpretability workflow or decision-support scenario to make the method’s impact more tangible?

---

> ### Author Response · Authors · 2025-11-22
> **Response to Reviewer 9y8U (1)**
>
> ## W1&Q1: Limited Methodological Contribution and Lack of In-Depth Comparison in the Literature Review
> ### Novelty of the interpretable paradigm
> Contrary to the reviewer’s comments, we assert that our work establishes a fundamentally new interpretability paradigm for visual prompt tuning (VPT). Specifically, we introduce a paradigm that integrates both attribution and concept-based interpretation to explain VPT—addressing a domain where previous methods relied exclusively on abstract, non-interpretable prompt embeddings.
>
> * **First interpretable paradigm for VPT**: Prior works in prompt tuning focus on learning abstract prompt tokens, which lack human-understandable meaning and offer limited transparency. Our framework is the first to propose learning interpretable visual prompts, representing a clear departure from existing approaches and enabling direct insight into the model’s decision-making process.
> * **Concept-attribution hybrid prompt interpretation**: We introduce a novel mechanism that enables direct semantic interpretation of prompts by associating visual prompts with explainable concepts grounded in specific image regions. Previous prototype-based methods can not by trivially applied to VPT due to the lack of concept-prompt linkage, cross-layer interpretation and class-agnostic concepts, which we will elaborate in the next section.
>
> ### Beyond simple extension to alignment techniques
> Conventional alignment techniques—such as ProtoPNet—cannot be trivially applied to visual prompt tuning (VPT) due to three key limitations:
>
> * **Lack of concept-prompt linkage:** Previous methods have not explored how to connect human-understandable concepts to the abstract prompts learned for adapting pre-trained ViT models. This linkage is essential for making prompt-based adaptation interpretable.
> * **Lack of cross-layer interpretation:** Existing approaches typically extract concepts only from the final layer of a neural network. They cannot interpret visual prompts learned at different ViT layers, nor do they capture cross-layer interactions among prompts. This is a critical gap, as VPT inherently learns prompts across multiple layers.
> * **Lack of class-agnostic concepts:** Prior methods generally interpret predictions for each category using separate sets of concepts, which limits their ability to analyze shared behaviors and generalize across classes.
>
> To bridge these gaps between alignment techniques and VPT, IVPT addresses all three limitations and proposes **multi-layer, category-agnostic concept**  prototypes that are **directly linked to visual prompts** and explicitly tied to image regions. A fine-to-coarse grouping mechanism with a concept region consistency loss aligns shallow and deep concepts into a coherent hierarchy, which prior work does not provide.
>
> ### Beyond existing visual prompt tuning
> Current efficient fine-tuning methods treat prompts as opaque embeddings, lacking (i) spatial grounding, (ii) human-understandable concept definition, and (iii) cross-layer semantic structure. Our framework maintains VPT's efficiency but redefines prompts as region-grounded, category-shared concepts with per-concept prediction scores. This shifts interpretability from final features/logits to the prompts themselves, an approach unaddressed in prior VPT literature.
>
> ### Detailed discussion of existing methods in related works
> We have provided a detailed analysis of the limitations of previous interpretation and VPT methods in the previous sections of this response. We have substantially expanded the Related Works section in the revised manuscript based on these analysis, to provide a deeper and more critical discussion of the limitations of existing interpretability methods and their applicability to VPT. For each related work, we now include a concise summary of its limitations, highlighting why these approaches cannot be trivially adapted to interpret VPT:
>
> * **Concept-based methods (e.g., CRAFT, TCAV):** These methods focus on high-level abstractions but typically operate at a single layer and lack region-level grounding. They do not provide mechanisms for linking concepts to prompt embeddings or for multi-layer interpretability.
> * **Attribution-based methods (e.g., Grad-CAM, Deep Visualization):** While effective at identifying influential regions, these approaches do not offer semantic concept alignment or interpretability of prompt tokens, especially across multiple layers.
> * **Part-prototype networks (e.g., ProtoPNet, TesNet, Huang et al.):** These frameworks interpret features at the final layer and are generally restricted to class-specific prototypes. They lack the ability to model cross-layer interactions, concept-prompt linkage, and category-agnostic concepts essential for VPT.
> * **Visual Prompt Tuning methods (e.g., VPT, E²VPT):** Existing VPT approaches focus on parameter-efficient adaptation but rely on abstract prompt embeddings, offering limited transparency and interpretability.

---

> ### Author Response · Authors · 2025-11-22
> **Response to Reviewer 9y8U (2)**
>
> ## W2&Q2: Potential Use Case and Concrete Application Setting
> We believe IVPT’s strong interpretability could open up a wide range of practical applications. Below, we provide two concrete examples to illustrate how our concept alignment framework can assist in real-world scenarios.
>
> ### Task Understanding
> Our experiment on the Gleason-2019 pathology dataset demonstrates how IVPT can enhance task understanding and model transparency in medical applications. As shown in Figure 4(a), IVPT automatically discovers concept prototypes corresponding to meaningful histological patterns, such as glandular lumen (green regions) and diseased glandular vesicles (purple regions). Furthermore, Figure 7 in Appendix F presents per-concept importance scores, revealing how different concepts contribute to each Gleason grade. This enables clinicians to inspect which tissue patterns the model relies on when assigning a grade and to verify whether these patterns are clinically plausible—facilitating more informed and trustworthy deployment in healthcare settings.
>
> ### Model Debugging & Bias Detection
> IVPT’s concept-level scores and region maps make it straightforward to identify which concepts the model relies on when making predictions, especially in cases of misclassification. This interpretability facilitates the detection of problematic patterns which can lead to bias or unreliable predictions. By examining misclassified examples in fine-grained recognition tasks, IVPT enables us to pinpoint instances where the prediction is dominated by a spurious concept rather than on discriminative object features. Such insights allow researchers to construct targeted bad cases for data curation, refine model training strategies, and systematically address underlying sources of bias. To further illustrate these practical benefits, we will add a dedicated section in the Appendix G.3 of the revised manuscript describing these cases in detail, demonstrating how IVPT can assist in model debugging and the mitigation of bias in downstream tasks.
>
> ## Q3: Generalizability Across VLM Architectures
> IVPT is designed to be broadly compatible with any ViT-style visual encoder, as it operates solely on patch tokens and attention blocks without relying on architecture-specific components. This makes our approach inherently flexible and adaptable to a wide range of backbone designs.
>
> To further evaluate generalization in vision–language model (VLM) settings, we instantiate IVPT on both CLIP (ViT-B/16, ViT-L/14) and SigLIP (Base, Large) architectures. These models share the ViT-style visual interface but differ in training objectives and text tower configurations. In our experiments, we attach multi-layer concept prototypes and prompts to the frozen visual encoder, leaving the text tower unchanged and using it as classifier weights.
>
> As summarized in the table below, IVPT achieves highly consistent results across four different VLM backbones (CLIP ViT-B/16, CLIP ViT-L/14, SigLIP-Base, SigLIP-Large). Notably, CLIP-based models tend to yield slightly higher consistency, while SigLIP-based models offer marginally better stability and accuracy; however, these differences are minor. These results demonstrate that IVPT’s alignment mechanism generalizes robustly across diverse VLM architectures and training objectives, without dependence on any particular backbone design.
>
> | Models | Con. | Sta. | Acc. |
> | - | - | - | - |
> | CLIP (ViT-B/16)     | 78.6     | 75.3     | 83.3     |
> | CLIP (ViT-L/14)     | 78.2     | 79.2     | 84.5     |
> | SigLIP (Base)     | 76.3     | 74.2     | 83.5     |
> | SigLIP (Large)     | 75.3     | 77.2     | 85.1     |
>
> ## Q4: Using Concept Alignment in Interpretability Workflows
> To illustrate the practical impact of IVPT’s concept alignment, we have presented a concrete example in Section 4.3 and Figure 4(a) in the original manuscript, in the context of a prostate cancer grading decision-support workflow using the Gleason-2019 pathology dataset.
>
> **Workflow Steps:**
> 1) **Concept Region Mapping:** For each biopsy patch, IVPT generates concept region maps that highlight distinct tissue patterns, such as glandular lumen and diseased vesicles.
> 2) **Per-Concept Importance Scoring:** IVPT computes importance scores for each concept, indicating how much each tissue pattern contributes to the predicted Gleason grade.
> 3) **Clinician Inspection and Verification:** The clinician can inspect the top-ranked concepts and their spatial locations within the biopsy image. This enables verification that the model is relying on clinically relevant patterns (e.g., glandular architecture) rather than artifacts or background regions.
> 4) **Actionable Review and Data Curation:** Cases where high-confidence predictions are driven by suspicious or non-clinical concepts can be flagged for further review. These flagged cases can inform targeted data curation or model refinement, improving reliability and trustworthiness.

---

### Author Response · Authors · 2025-12-01
**Rebuttal Summary for AC**

We provide a summary of our rebuttal here for AC to better understand the key aspects of our response. Overall, we have received three positive scores and one marginally negative score from the reviewers. In our rebuttal, we have comprehensively **addressed all concerns** raised by the reviewers, significantly strengthening the paper's contribution and clarifying its novelty, applicability, and methodological soundness.

Reviewer PdEH, the only reviewer who gave a score of 4 (marginally below acceptance), raised two main concerns. Our rebuttal addresses these as follows:
* **Concern 1:** The need for clearer motivation on why IVPT can explain domain shift.
    * In our response, we provide a detailed explanation that, contrary to reviewer's belief, interpretable prompts in **IVPT is able to reveal how model adapts under domain shift** by showing which concepts emerge or change in importance during fine-tuning.
    * In the revised paper, additional qualitative results on the VLCS dataset that further illustrate how semantic priorities shift across domains, making domain adaptation interpretable.

* **Concern 2:** A misunderstanding that our interpretable prompts only partially contribute to the final logits.
    * In the response, we clarify this misunderstanding by emphasizing that the final classification logits are computed **exclusively from the interpretable prompt tokens**. The image patch tokens from the frozen backbone **do not** contribute directly to the logits.


We are encouraged that all the other three reviewers recognized the value of our contribution, scoring it positively, and we have successfully addressed their specific points.

Responses of main concerns from Reviewer 9y8U are
* **Concern 1:** IVPT represents an extension of alignment techniques
    * We clarified that IVPT is not only a technical modification but introduces the first interpretability paradigm in Visual Prompt Tuning, bridging critical gaps left by prior non-VPT methods.
    *  We substantially expanded the Related Works section in the revised paper with a deeper, more critical discussion with existing works.
* **Concern 2:** Require concrete use cases on interpretable method.
  * In response and revised paper, we provided concrete use cases of IVPT in model debugging and bias detection .

Main concerns from Reviewer 9y8U are the stability of the learned concept prototypes to initialization and domain shift. We addressed this by running additional experiments with multiple random seeds, showing low performance variance, and demonstrating that our prototypes generalize to novel, unseen classes within the dataset.

For Reviewer RFsS, the main concern was improving experimental consistency by providing evaluation metrics across datasets and adding more qualitative results for complex datasets. We directly addressed this by incorporating predictive performance metrics and providing additional qualitative visualizations for PartImageNet and PASCAL-Part, as requested by the reviewer.

To conclude, we have successfully demonstrated that our work establishes a fundamentally new interpretability paradigm for Visual Prompt Tuning (VPT). The revisions, guided by the reviewers' feedback, have been comprehensively incorporated into the manuscript (highlighted in blue). These concrete improvements include: a substantially expanded Related Works section, new quantitative results and qualitative analyses on broader datasets, a dedicated section illustrating practical applications, and rigorous empirical validation of our method's faithfulness. This extensive new evidence effectively counters the initial skepticism and underscores our work's unique role in bridging the critical gap between interpretability and visual prompt tuning.

---

### Meta-Review · Area_Chair_EWTs · 2025-12-24

**Summary:**

The current paper explores the idea of using visual prompt tuning in the context of a prototype-based interpretable architecture. The novelty of this work compared to other prototype-based interpretability methods lie in the integration of cross-layer concept prototypes with visual prompt tuning, and cross-layer aggregation of features from these prototypes to generate interpretable prompts at different network depths.
Empirical evaluation demonstrate the capability of the proposed architecture to recover interpretable visual prompts while preserving accuracy, beating other prototype-based methods on fine-grained benchmarks, particularly CUB-200-2011 and pathology datasets.

Among the strengths of the paper, reviewers appreciated the clear motivation and intent to position the work within the broader context of interpretable machine learning, the comprehensive empirical analysis which includes multiple datasets and insightful visualizations, and the generalizability of the proposed method across transformer architectures.

The main weakness raised in the reviews have to do with the novelty of the methodological contribution.
Reviewers noted that the approach relies heavily on established techniques, and that the current presentation does not sufficiently situate the method against relevant prior work paper to show what is actually conceptually new.
In particular, the current work does not cite and discuss very relevant previous work making use of similar mechanisms for interpretability also in transformer networks, like Hong et al. "Concept-Centric Transformers" WACV 2024 and Rigotti et al. "Attention-based Interpretability with Concept Transformers" ICLR 2022. Both establish precedents for linking image patches to interpretable prototypes via attention weights for concept discovery and interpretability over learned tokens and should be discussed in the related work section, and properly differentiated against. Both models also achieve comparable results on CUB-200-2011 which should also be discussed in the empirical evaluation to contextualize the current contribution.

The reviewing panel therefore requests that authors make sure that the resubmitted camera-ready version of the paper will include proper citations and positioning with respect to these works and related attention-based interpretability models, as such an in-depth comparative analysis  will be essential to counter the novelty concerns raised in the reviews.

**Reviewer Concerns:**

* Addressed in the rebuttal:
  - concerns about differentiation with respect to existing prototype-based interpretability methods have been addressed in parts in the rebuttal by clarifying the novelty of the proposed cross-layer concept prototypes and their integration with visual prompt tuning
  - Questions about transferability of the method across architectures have been addressed by demonstrating consistent results across multiple models like CLIP and SigLIP

* Not addressed in the rebuttal:
  - concerns about limited innovation: reviewers noted that the method primarily combines existing techniques (visual prompt tuning, prototype-based interpretability) without introducing fundamentally new interpretability paradigms
  - Experiments hare limited to image classification tasks

**Reviewer Scores:**

| Reviewer | initial score | predicted final score |
|---:|---:|---:|
| 9y8U | 6 | 6 |
| ANRv | 6 | 6 |
| PdEH | 4 | 4 |
| RFsS | 6 | 6 |

---

### Decision · Program_Chairs · 2026-01-26

Accept (Poster)